# Multi-Head Attention as a Source of Catastrophic Forgetting in MoE Transformers

Anrui Chen [1]  Ruijun Huang [1 2]  Xin Zhang [1]  Fang Dong [1]  Hengjie Cao [1 3]  Zhendong Huang [1]  Yifeng Yang [1]
Mengyi Chen [1]  Jixian Zhou [1]  Mingzhi Dong [4]  Yujiang Wang [5 6]  Jinlong Hou [3]  Qin Lv [7]  Robert P. Dick [8]
Yuan Cheng [3]  Tun Lu [1]  Fan Yang [1]  Li Shang [1 9]

## Abstract

Mixture-of-Experts (MoE) architectures are appealing for continual learning because sparse routing should localize updates and reduce interference, yet MoE Transformers still forget substantially even with sparse, well-balanced expert utilization. We attribute this gap to a pre-routing bottleneck: multi-head attention concatenates head-specific signals into a single post-attention router input, forcing routing to act on co-occurring feature compositions rather than separable head channels. We show that this router input simultaneously encodes multiple separately decodable semantic and structural factors with uneven head support, and that different feature compositions induce weakly aligned parameter-gradient directions; as a result, routing maps many distinct compositions to the same route. We quantify this collision effect via a route-wise effective composition number $N_{\text{eff}}$ and find that higher $N_{\text{eff}}$ is associated with larger old-task loss increases after continual training. Motivated by these findings, we propose MH-MoE, which performs headwise routing over sub-representations to increase routing granularity and reduce composition collisions. On TRACE across multiple backbones, MH-MoE consistently improves the retention–accuracy trade-off over LoRA-MoE variants.

## 1. Introduction

Mixture-of-Experts (MoE) architectures (Jacobs et al., 1991; Jordan & Jacobs, 1994) are appealing for continual and multi-task learning because routing can localize updates to a subset of experts, potentially reducing gradient interference and thereby mitigating catastrophic forgetting (Chen et al., 2023; Kang et al., 2026; Li et al., 2024b; 2025a; Dou et al., 2024). In principle, expert specialization should preserve previously learned knowledge by isolating task-specific parameter changes (Ramasesh et al., 2021; Davari et al., 2022; Goodfellow et al., 2013).

Despite this promise, our empirical results show that MoE Transformers continue to suffer substantial catastrophic forgetting, even when expert utilization is sparse and well-balanced. On TRACE, the MoE baseline attains markedly negative backward transfer (BWT $= -11.0\%$ on Ministral-3-8B; Table 1), reflecting systematic degradation on earlier tasks as later tasks are learned. Expert modularity alone is therefore insufficient to prevent interference. A natural question follows: if experts are intended to isolate knowledge, where does task interference actually arise in MoE Transformers?

In this work, we argue that catastrophic forgetting in MoE Transformers is primarily caused by a structural bottleneck introduced before expert routing: multi-head attention. In standard Transformer architectures, outputs from multiple representation heads, each encoding distinct and heterogeneous features, are concatenated into a single vector prior to routing. This design implicitly assumes that the concatenated representation forms a coherent feature space suitable for expert selection.

Our analysis shows that this assumption is generally violated: multi-head attention produces a post-attention router input in which multiple feature signals co-occur in a single vector, while their support is unevenly distributed across heads. As a result, MoE routing must make expert decisions based on feature co-occurrence, which is prone to composition collisions and interference. Concretely, we establish three findings:

[1]Fudan University, Shanghai, China [2]Greater Bay Area National Center of Technology Innovation, Research Institute of Tsinghua University in Shenzhen, Shenzhen, China [3]Shanghai Innovation Institute, Shanghai, China [4]University of Bath, Bath, United Kingdom [5]Department of Engineering Science, University of Oxford, Oxford, UK [6]Oxford Suzhou Centre for Advanced Research, University of Oxford, Suzhou, China [7]Department of Computer Science, University of Colorado Boulder, Colorado, USA [8]Department of Electrical Engineering and Computer Science, University of Michigan [9]Shenzhen Loop Area Institute, Shenzhen, China. Correspondence to: Li Shang <lishang@fudan.edu.cn>.

*Proceedings of the 43rd International Conference on Machine Learning*, Seoul, South Korea. PMLR 306, 2026. Copyright 2026 by the author(s).

**Multi-head attention mixes head-structured feature signals.** The post-attention router input jointly encodes multiple separately decodable semantic and structural features (Fig. 1), while their support is highly non-uniform across representation heads (Fig. 2), showing that head-specific feature channels are aggregated into a single vector before routing.

**Feature compositions induce diverse learning signals.** Different feature compositions produce composition-conditioned parameter-gradient directions with low cosine agreement (Fig. 3), implying that a single shared update direction cannot align well with compositions simultaneously.

**Composition collisions under MoE routing amplify forgetting.** Because MoE routing compresses the multiplexed router input into a single expert decision, distinct feature compositions collide on the same route (high route-wise effective composition number $N_{\text{eff}}$; Fig. 4). Routes with higher $N_{\text{eff}}$ exhibit larger old-task loss increases after continual training, linking composition mixing to catastrophic forgetting (Fig. 5).

Motivated by these findings, we propose MH-MoE, which performs routing independently across multiple representation heads rather than making a single expert decision from a head-mixed router input. This head-wise routing yields two practical benefits:

*Mitigates forgetting with minimal accuracy loss.* Head-wise routing reduces feature-composition collisions within each update destination (lower route-wise $N_{\text{eff}}$). Because different compositions induce weakly aligned learning signals, this reduces gradient conflict and improves retention.

*Task-agnostic and streamable.* MH-MoE does not rely on task boundaries, task IDs, or replay buffers, and uses the same token-level routing mechanism during training and inference, making it naturally compatible with continuous streams where task switches are unknown.

We evaluate MH-MoE on TRACE (8 tasks) with pretrained backbones, comparing against a standard MoE baseline (LoRAMoE). MH-MoE consistently improves the retention–accuracy tradeoff: on Llama-3.2-3B, OP increases from 46.0 to 52.2 and BWT improves from $-7.1$ to $-2.2$; on Ministral-3-8B, OP increases from 52.7 to 58.6 and BWT improves from $-11.0$ to $-1.7$.

## 2. Analysis

In this section, we explain why MoE Transformers can still suffer substantial catastrophic forgetting. All analyses use Qwen3-0.6B (and its MoE variant) on C-STANCE and FOMC datasets.

### 2.1. Post-Attention Router Inputs Are Head-Mixed and Multi-feature

This subsection asks whether the router input provides separable signals that routing can exploit, or instead mixes multiple factors so that routing must act on their co-occurrence. We answer this with two analyses: (i) we test which semantic/structural variables are linearly decodable from the post-attention representation and whether their probe-induced subspaces overlap; (ii) we quantify whether these signals are supported unevenly across representation heads.

**Features correspond to linearly decodable signals in the representation.** Let $h_t^{(\ell)} \in \mathbb{R}^d$ denote the post-attention token representation at position $t$ in layer $\ell$, which serves as the router input in MoE layers. We define a *feature $Y$* as a discrete variable whose value is linearly decodable from $h_t^{(\ell)}$. For each feature $Y$ and layer $\ell$, we train a multinomial linear probe on frozen representations:

$$\hat{p}_\ell(y \mid h) = \text{Softmax}\left(W_Y^{(\ell)} h + b_Y^{(\ell)}\right). \qquad (1)$$

The probe induces a feature-specific decoding geometry via the *decoding subspace*

$$\mathcal{S}_Y^{(\ell)} = \text{span}\left((W_Y^{(\ell)})^\top\right) \subseteq \mathbb{R}^d, \qquad (2)$$

which captures the linear directions in $h_t^{(\ell)}$ predictive of $Y$. We instantiate $Y$ using salient semantic and structural variables: domain identity, stance label, token-frequency bucket, and relative-position bucket.

**Multiple features are mixed in a single router input.**

All studied variables are predicted substantially above chance across layers (Fig. 1a), showing that the same router input $h_t^{(\ell)}$ simultaneously carries semantic (domain/stance) and structural (frequency/position) information. Moreover, probe-induced decoding subspaces have small pairwise overlap (Fig. 1b), indicating that these signals occupy largely distinct linear directions within $h_t^{(\ell)}$. However, MoE routing applies a single scoring function to the full vector $h_t^{(\ell)}$ and collapses these co-existing directions into one expert decision. As a result, the routing score reflects their joint configuration, so tokens that differ along one feature direction can still be mapped to the same route when other directions dominate. Together, $h_t^{(\ell)}$ is *multiplexed*: multiple linearly decodable signals co-exist in one vector, and a single routing score computed from $h_t^{(\ell)}$ must implicitly trade off among them when they co-occur.

**Features are head-structured.** The multiplexing in $h_t^{(\ell)}$ is not uniform across representation heads. For each feature $Y$, we quantify head-wise *ablation-based importance*: we remove one head at the ablation site and measure how much

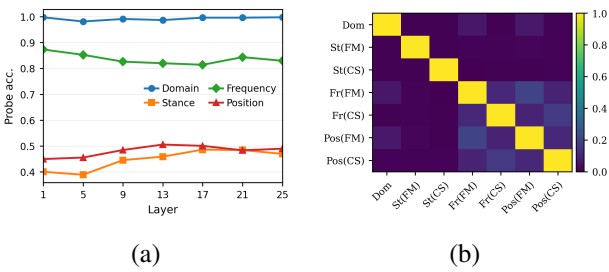

Figure 1. **The router input multiplexes multiple decodable features.** (a) Linear probes on post-attention states $h_t^{(\ell)}$ predict domain/stance (semantic) and frequency/position (structural) well above chance across layers. (b) Probe-induced decoding subspaces have small overlap. Abbreviations: CS=C-STANCE, FM=FOMC, Dom=Domain, St=Stance, Fr=Frequency, Pos=Position.

the feature's probe performance degrades on the router input (Appendix A.1). Normalizing these importance scores across heads yields a per-feature distribution over heads (*shares*) that summarizes where the decodable signal is concentrated. We observe highly non-uniform head–feature patterns (Fig. 2): for each feature, a small subset of heads accounts for a disproportionate share of importance, and the dominant head subsets differ across features.

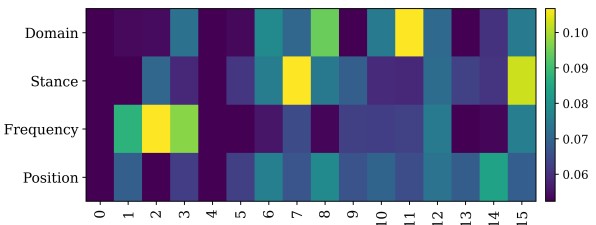

Figure 2. **Feature signals are head-structured but mixed in the router input.** For each feature $Y$, we ablate one head at a time and measure the drop in probe accuracy on $h_t^{(\ell)}$ as its causal importance. Importance is sharply concentrated on a few heads and varies across features.

These results show that the standard router input $h_t^{(\ell)}$ is both *multiplexed* and *head-structured*: multiple separable feature signals are present, but their support is uneven across heads and then aggregated into a single vector. Standard MoE routing compresses $h_t^{(\ell)}$ into a single expert decision, so it cannot preserve head-wise separation when multiple signals co-occur. Consequently, tokens with different feature compositions can map to the same route, forcing parameter sharing across heterogeneous learning signals.

## 2.2. Feature Compositions Induce Distinct Learning Signals

This subsection tests whether different feature compositions induce distinct parameter-update directions by comparing composition-conditioned gradients.

**Feature compositions.** Let $\mathcal{Y}_1, \ldots, \mathcal{Y}_m$ be the label spaces for $m$ features decodable from $h_t^{(\ell)}$. We define the *feature composition* of a token representation $h_t^{(\ell)}(x)$ as the tuple

$$c\Big(h_t^{(\ell)}(x)\Big) = (y_1, \ldots, y_m), \qquad y_i \in \mathcal{Y}_i. \quad (3)$$

Although the full product $\mathcal{Y}_1 \times \cdots \times \mathcal{Y}_m$ can be large, we work with the empirical subset observed in data. In our experiments, $(y_1, \ldots, y_m)$ is instantiated using ground-truth labels when available (domain/stance) and bucketed statistics (frequency/position).

**Gradients as learning signals.** Let $\ell_{x,t}(\theta) = -\log p_\theta(x_{t+1} \mid x_{\leq t})$ denote the token-level next-token loss. For a parameter block $\theta^{(\ell)}$ at layer $\ell$ that is updated during continual training, we define the token-level parameter-gradient

$$g_{x,t}^{(\ell)} = \nabla_{\theta^{(\ell)}} \ell_{x,t}(\theta) \in \mathbb{R}^{|\theta^{(\ell)}|}. \quad (4)$$

Here $|\theta^{(\ell)}|$ denotes the number of scalar parameters in the block $\theta^{(\ell)}$.

**Composition-conditioned mean directions.** For a composition $c$, let

$$\mathcal{S}_c^{(\ell)} = \{(x,t) \,:\, c(h_t^{(\ell)}(x)) = c\}$$

be the set of tokens with composition $c$ at layer $\ell$. We aggregate per-token gradients into a composition-conditioned mean direction:

$$\bar{g}^{(\ell)}(c) = \frac{1}{|\mathcal{S}_c^{(\ell)}|} \sum_{(x,t)\in\mathcal{S}_c^{(\ell)}} \frac{g_{x,t}^{(\ell)}}{\|g_{x,t}^{(\ell)}\|_2 + \varepsilon}, \quad (5)$$

where $\varepsilon > 0$ is a small constant for numerical stability. Normalizing each token gradient focuses this analysis on directional agreement (interference/alignment) rather than magnitude. We compare compositions via cosine similarity

$$\text{Sim}^{(\ell)}(c_1, c_2) = \cos\Big(\bar{g}^{(\ell)}(c_1), \bar{g}^{(\ell)}(c_2)\Big). \quad (6)$$

**Within-composition coherence vs. cross-composition weak alignment.** We evaluate two notions of gradient-direction agreement: (i) **within-composition coherence**: for each composition $c$, randomly partition $\mathcal{S}_c^{(\ell)}$ into two disjoint subsets $A$ and $B$, compute $\bar{g}_A^{(\ell)}(c)$ and $\bar{g}_B^{(\ell)}(c)$ via Eq. (5), and measure $\cos(\bar{g}_A^{(\ell)}(c), \bar{g}_B^{(\ell)}(c))$; (ii) **cross-composition agreement**: sample distinct compositions $c_1 \neq c_2$ and compute $\text{Sim}^{(\ell)}(c_1, c_2)$. Fig. 3 shows that within-composition directions are consistently aligned, while cross-composition similarities concentrate near zero. Thus, different feature compositions induce diverse learning signals: a single shared update direction cannot simultaneously align well with many compositions. This observation

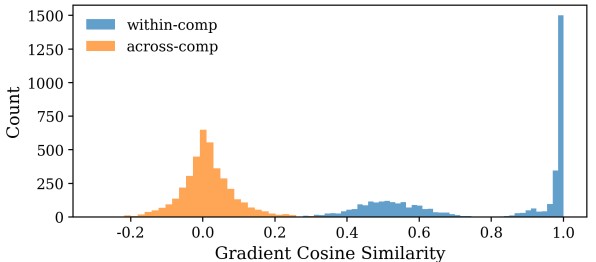

*Figure 3.* **Different feature compositions induce distinct gradient directions.** Histogram of cosine similarity between composition-conditioned mean gradient directions (Eq. (5)–(6)). Within-composition splits are strongly aligned, while different compositions cluster near zero, indicating weak cross-composition agreement.

motivates why composition mixing within a single MoE route can be harmful.

These results establish that feature compositions correspond to stable, composition-specific update directions: gradients are coherent within a composition but weakly aligned across compositions. Consequently, when multiple compositions share parameters, their updates are more likely to interfere.

### 2.3. From Composition Mixing to Forgetting in MoE Routing

This subsection asks whether *composition mixing* induced by standard MoE routing predicts catastrophic forgetting. We answer this in three steps: (i) define route-wise composition mixing under the old-task token distribution; (ii) define route-conditioned old-task loss and route-wise forgetting, and derive a theoretical link between higher mixing and greater susceptibility; (iii) empirically test how forgetting varies with mixing across routes while controlling for old-task exposure.

**Route assignment.** For clarity, we present the analysis under top-1 routing; top-$k$ follows analogously. Consider an MoE layer with $K$ experts. Given router input $h_t^{(\ell)}(x) \in \mathbb{R}^d$, the router produces logits $a_t^{(\ell)}(x) \in \mathbb{R}^K$ and selects

$$r_t^{(\ell)}(x) = \arg\max_{k \in [K]} a_{t,k}^{(\ell)}(x), \qquad (7)$$

which we call the *route* for token $(x, t)$ at layer $\ell$.

**Route-wise composition mixing under old-task tokens.** Let $c(h_t^{(\ell)}(x))$ denote the feature composition (Eq. (3)). We define the distribution of compositions conditioned on route $r$ under the old-task token distribution $\mathcal{D}_{\mathrm{old}}$:

$$p^{(\ell)}(c \mid r) = \Pr_{(x,t) \sim \mathcal{D}_{\mathrm{old}}} \left[ c(h_t^{(\ell)}(x)) = c \,\Big|\, r_t^{(\ell)}(x) = r \right]. \qquad (8)$$

If routing were composition-selective, $p^{(\ell)}(c \mid r)$ would concentrate on a small number of compositions. A broad

$p^{(\ell)}(c \mid r)$ indicates *composition mixing* within the route. We focus on $\mathcal{D}_{\mathrm{old}}$ because forgetting is evaluated on old-task tokens: higher mixing implies that a larger and more diverse fraction of old-task compositions share a route whose parameters will later be updated by new-task training, increasing the chance of interference.

**Effective composition number.** We quantify mixing using the effective number of compositions:

$$N_{\mathrm{eff}}^{(\ell)}(r) = \frac{1}{\sum_c \left( p^{(\ell)}(c \mid r) \right)^2}. \qquad (9)$$

This equals $1$ if all tokens routed to $r$ share the same composition, and increases as $p^{(\ell)}(c \mid r)$ spreads.

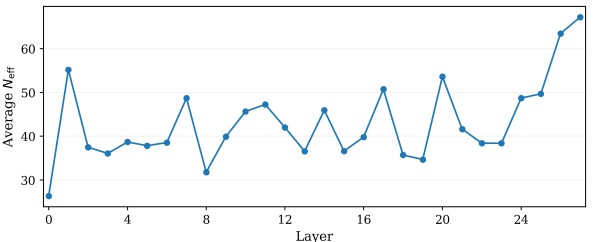

*Figure 4.* **Composition mixing persists across layers.** Old-task mass-weighted average effective composition number $N_{\mathrm{eff}}$ (Eq. (9)) across routes in each MoE layer. Values well above $1$ show that routes aggregate multiple feature compositions under the old-task distribution.

Fig. 4 shows that the old-task exposure-weighted $N_{\mathrm{eff}}$ remains substantially above $1$ throughout the MoE stack, indicating that routes typically aggregate multiple feature compositions under the old-task token distribution.

**Route-conditioned old-task loss and forgetting.** For each MoE module at layer $\ell$, we compute the route-conditioned loss on old-task data:

$$L_{\mathrm{old}}^{(\ell)}(r; \theta) = \mathbb{E}\left[ -\log p_\theta(x_{t+1} \mid x_{\leq t}) \,\Big|\, r_{x,t}^{(\ell)} = r \right], \quad (10)$$

and define route-wise forgetting as

$$\Delta L_{\mathrm{old}}^{(\ell)}(r) = L_{\mathrm{old}}^{(\ell)}(r; \theta_{\mathrm{new}}) - L_{\mathrm{old}}^{(\ell)}(r; \theta_{\mathrm{old}}). \qquad (11)$$

Here $\theta_{\mathrm{old}}$ is the model parameter after training on old tasks, and $\theta_{\mathrm{new}}$ is the parameter after subsequent continual training on new tasks.

**Why higher mixing increases susceptibility.** Our key mechanism is: if a route $r$ mixes many distinct compositions, then no small subset of "well-protected" compositions can account for most old-task tokens routed to $r$. Consequently, a nontrivial fraction of the old-task mass must lie in compositions that are not reliably protected under typical update directions, making the route more susceptible to forgetting. This is consistent with Fig. 3: since cross-composition gradient directions are weakly aligned, an update direction

that preserves a small subset of compositions is unlikely to simultaneously preserve many others that share the same route.

We formalize this in two steps. Lemma 2.1 relates the effective composition number $N_{\text{eff}}(r)$ to how much probability mass can be concentrated on any $m$ compositions. Theorem 2.2 then converts this mass guarantee into a lower bound on the route-level old-loss increase.

**Lemma 2.1** (Mixing mass bound). *Fix a route $r$ with composition distribution $p(c \mid r)$ over $c \in \mathcal{C}$, and define*

$$N_{\text{eff}}(r) \;=\; \left( \sum_{c \in \mathcal{C}} p(c \mid r)^2 \right)^{-1}.$$

*Then for any $S \subseteq \mathcal{C}$ with $|S| \leq m$,*

$$\Pr_{C \sim p(\cdot|r)}[C \notin S] \;\geq\; 1 - \sqrt{\frac{m}{N_{\text{eff}}(r)}}.$$

**Theorem 2.2** (Composition mixing increases forgetting susceptibility). *Fix a route $r$. For each $c \in \mathcal{C}$, let $F_c(\theta)$ be the old-task loss restricted to tokens routed to $r$ with composition $c$, and define*

$$F_r(\theta) \;=\; \mathbb{E}_{C \sim p(\cdot|r)}\big[F_C(\theta)\big].$$

*Consider one update $\theta^+ = \theta - \eta\hat{u}$ with $\|\hat{u}\| = 1$, $\eta > 0$. Assume each $F_c$ is $L$-smooth and $\|\nabla F_c(\theta)\| \leq G$ for all $c$.*

*Let $S \subseteq \mathcal{C}$ with $|S| \leq m$ denote a subset of compositions that happen to be well-aligned with the update. Suppose there exist $\rho \in (0, 1]$ and $\kappa > 0$ such that for all $c \notin S$,*

$$\Pr\big(F_c(\theta^+) - F_c(\theta) \;\geq\; \kappa\big) \;\geq\; \rho,$$

*where the probability is over the randomness defining $\hat{u}$. Let $a := \sqrt{m/N_{\text{eff}}(r)}$. Then*

$$\mathbb{E}\big[F_r(\theta^+) - F_r(\theta)\big] \;\geq\; (1 - a)_+ B_{\text{out}} \;-\; a\,B_{\text{in}},$$

*where*

$$B_{\text{out}} = \rho\kappa - (1 - \rho)\Big(\eta G + \tfrac{L\eta^2}{2}\Big), \quad B_{\text{in}} = \eta G + \tfrac{L\eta^2}{2}.$$

*In particular, since $a$ decreases with $N_{\text{eff}}(r)$, the lower bound is nondecreasing in $N_{\text{eff}}(r)$ whenever $B_{\text{out}} + B_{\text{in}} > 0$; and it becomes positive once $N_{\text{eff}}(r)$ is large enough and $B_{\text{out}} > 0$.*

Proofs in Appendix A.2.

**Empirical association between mixing and forgetting.** We empirically test the theorem's qualitative prediction by pooling routes across modules and examining how $\Delta L_{\text{old}}$ varies with the mixing score $N_{\text{eff}}$. To ensure the trend reflects exposure of *old-task tokens* (rather than being dominated by many rarely used routes), we form bins

by *mass-quantiles* of $N_{\text{eff}}$ under the old-task routing mass $\text{mass}_{\text{old}}(r) = \Pr_{(x,t) \sim \mathcal{D}_{\text{old}}}[r_{x,t}^{(\ell)} = r]$, so each bin contains approximately equal total old-task routing mass. Within each bin, we report the mean (and standard error) of $\Delta L_{\text{old}}$ across routes. Fig. 5 shows that routes with larger $N_{\text{eff}}$ are associated with greater $\Delta L_{\text{old}}$, consistent with the susceptibility mechanism highlighted by Theorem 2.2.

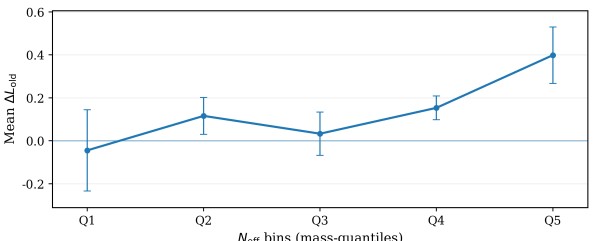

*Figure 5.* **More mixed routes forget more.** Route-wise old-task loss increase $\Delta L_{\text{old}}$ versus effective composition number $N_{\text{eff}}$ (Eq. (9)). Routes are binned by mass-quantiles of $N_{\text{eff}}$ under old-task routing exposure $\text{mass}_{\text{old}}(r)$, so each bin contains comparable old-task token mass. Points report mean $\Delta L_{\text{old}}$ with standard error, showing a positive association between mixing and forgetting.

# 3. Multi-Head Mixture-of-Experts (MH-MoE)

We introduce **MH-MoE**, a Transformer–MoE layer that performs expert routing *independently over multiple sub-representations* of the post-attention state. Unlike standard MoE, which collapses the full representation into a single routing decision, MH-MoE factorizes the representation into $H$ head-aligned slices and routes each slice separately, yielding a tuple-valued routing decision. This increases routing resolution and reduces feature-composition collisions, improving retention in continual learning.

**Head-aligned splitting.** Let $h_t^{(\ell)} \in \mathbb{R}^d$ be the post-attention token representation at layer $\ell$ and position $t$. We partition the feature dimension into $H$ disjoint slices:

$$\begin{aligned} h_t^{(\ell)} &= \big[\, h_{t,1}^{(\ell)} \,\|\, \cdots \,\|\, h_{t,H}^{(\ell)} \,\big], \\ h_{t,m}^{(\ell)} &\in \mathbb{R}^{d/H}, \qquad m \in [H]. \end{aligned} \tag{12}$$

**Head-private routing.** Each head $m$ has its own router $W_{\text{rt}}^{(\ell,m)}$ and its own private expert bank $\{E_k^{(m)}\}_{k=1}^K$. The router selects the top-$k$ experts using only the head slice:

$$\begin{aligned} a_{t,m}^{(\ell)} &= W_{\text{rt}}^{(\ell,m)} h_{t,m}^{(\ell)} \in \mathbb{R}^K, \\ \mathcal{S}_{t,m}^{(\ell)} &= \text{TopK}(a_{t,m}^{(\ell)}, k) \subseteq [K], \\ \alpha_{t,m,j}^{(\ell)} &= \frac{\exp(a_{t,m,j}^{(\ell)})}{\sum_{q \in \mathcal{S}_{t,m}^{(\ell)}} \exp(a_{t,m,q}^{(\ell)})}, \qquad j \in \mathcal{S}_{t,m}^{(\ell)}. \end{aligned} \tag{13}$$

The overall routing decision is the tuple of selected expert sets $\mathbf{S}_t^{(\ell)} = \big(\mathcal{S}_{t,1}^{(\ell)}, \ldots, \mathcal{S}_{t,H}^{(\ell)}\big)$.

**Algorithm 1** MH-MoE

---

**Input:** token states $\{h_t^{(\ell)}\}_{t=1}^T$, $h_t^{(\ell)} \in \mathbb{R}^d$; heads $H$ ($H \mid d$); routers $\{W_{\mathrm{rt}}^{(\ell,m)}\}_{m=1}^H$; experts $\{E_j^{(m)}\}_{m=1,j=1}^{H,K}$; top-$k$.

**Output:** $\{y_t^{(\ell)}\}_{t=1}^T$, $y_t^{(\ell)} \in \mathbb{R}^d$.

**for** $t = 1$ **to** $T$ **do**

   Split $h_t^{(\ell)} = [h_{t,1}^{(\ell)} \| \cdots \| h_{t,H}^{(\ell)}]$, $h_{t,m}^{(\ell)} \in \mathbb{R}^{d/H}$; set $y_t^{(\ell)} \leftarrow \mathbf{0}$.

   **for** $m = 1$ **to** $H$ **do**

      $a_{t,m}^{(\ell)} \leftarrow W_{\mathrm{rt}}^{(\ell,m)} h_{t,m}^{(\ell)}$; $\mathcal{S}_{t,m}^{(\ell)} \leftarrow \mathrm{TopK}(a_{t,m}^{(\ell)}, k)$.

      $\alpha_{t,m,\cdot}^{(\ell)} \leftarrow \mathrm{Softmax}\left(a_{t,m,\cdot}^{(\ell)}\right)$ restricted to $\mathcal{S}_{t,m}^{(\ell)}$.

      $y_t^{(\ell)} \leftarrow y_t^{(\ell)} + \sum_{j \in \mathcal{S}_{t,m}^{(\ell)}} \alpha_{t,m,j}^{(\ell)} E_j^{(m)}\left(h_{t,m}^{(\ell)}\right)$.

   **end for**

**end for**

---

**Head-private expert output aggregation.** Each expert takes a $d/H$-dimensional slice and produces a full $d$-dimensional output:

$$E_k^{(m)} : \mathbb{R}^{d/H} \to \mathbb{R}^d. \tag{14}$$

Given $\mathbf{S}_t^{(\ell)}$, MH-MoE applies the selected experts for each head:

$$y_{t,m}^{(\ell)} = \sum_{j \in \mathcal{S}_{t,m}^{(\ell)}} \alpha_{t,m,j}^{(\ell)} E_j^{(m)}(h_{t,m}^{(\ell)}) \in \mathbb{R}^d, \qquad m \in [H], \tag{15}$$

and aggregates by summation:

$$y_t^{(\ell)} = \sum_{m=1}^H y_{t,m}^{(\ell)} \in \mathbb{R}^d. \tag{16}$$

**Implicit routing resolution.** Although MH-MoE stores only $HK$ experts (organized into $H$ private banks), the tuple $\mathbf{S}_t^{(\ell)}$ induces an implicit $\left(\binom{K}{k}\right)^H$-way partition of tokens. Equivalently, the selection-indexed composite mapping is

$$E_{\mathbf{S}}(h) = \sum_{m=1}^H \sum_{j \in \mathcal{S}_m} \alpha_{m,j} E_j^{(m)}(h_m). \tag{17}$$

where $h = [h_1 \| \cdots \| h_H]$, $\mathbf{S} = (\mathcal{S}_1, \ldots, \mathcal{S}_H)$.

## 4. Related Work

**Mixture of Experts.** Mixture of Experts (MoE) is a foundational paradigm for conditional computation, where a router dynamically selects among multiple experts so that different subsets of parameters specialize for different inputs (Jacobs et al., 1991; Jordan & Jacobs, 1994). Early work extended MoE beyond shallow mixtures to deep architectures with stacked routers and experts to increase capacity and expressivity (Eigen et al., 2013). A major practical breakthrough was the sparse MoE layer (Shazeer et al., 2017), which enforces sparse expert activation per example/token, improving scalability and training stability while reducing compute. Since then, MoE has been integrated into a wide range of neural backbones—including convolutional and Transformer-based models—and has achieved strong results across tasks. In the LLM regime, MoE is widely adopted to scale model capacity under fixed compute budgets, and substantial effort has been devoted to routing design (Lepikhin et al., 2021; Fedus et al., 2022; Du et al., 2022). Representative strategies include token-driven routing where each token selects its top-k experts (Shazeer et al., 2017; Fedus et al., 2022), expert-driven routing where experts select the top-k tokens to process (Zhou et al., 2022), and global assignment schemes that decide expert allocation at a higher granularity (Lewis et al., 2021; Riquelme et al., 2021).

**Catastrophic Forgetting.** Catastrophic forgetting refers to the rapid degradation of previously learned capabilities when a model is trained sequentially on new data, a phenomenon classically attributed to *parameter interference* in shared networks where updates for new tasks overwrite weights supporting earlier tasks (McCloskey & Cohen, 1989). Mitigation strategies broadly fall into (i) *regularization/importance-based* methods that constrain changes to parameters deemed crucial for past tasks, such as EWC (Kirkpatrick et al., 2017) and Synaptic Intelligence (Zenke et al., 2017); (ii) *replay and constraint-based* methods that preserve past behavior through episodic memory or gradient projection (Lopez-Paz & Ranzato, 2017; Rebuffi et al., 2017; He et al., 2026); and (iii) *architectural isolation/expansion* approaches that allocate different tasks to different parameter subsets or grow capacity over time to reduce interference. In the LLM setting, recent empirical studies show that continual instruction tuning and sequential domain adaptation can induce substantial forgetting across knowledge and reasoning abilities (Luo et al., 2025; Zixuan et al., 2023), motivating renewed study of continual learning under the scale and representation entanglement of modern LLMs (Shi et al., 2025). A complementary direction addresses stability–plasticity through external memory evolution in language agents (Tian et al., 2025); in contrast, our work focuses on parametric continual learning in MoE Transformers and identifies routing-induced composition collision as an internal source of forgetting.

## 5. Experiments

### 5.1. Experimental Setup

We evaluate whether MH-MoE improves continual learning by reducing forgetting while maintaining strong overall accuracy.

*Table 1.* **Continual learning performance on TRACE after training on all tasks.** We report final score on each dataset, Overall Performance (OP), and Backward Transfer (BWT). Abbreviations: CS=C-STANCE, FM=FOMC, MB=MeetingBank, PY=Py150, SQ=ScienceQA, NC=NumGLUE-cm, ND=NumGLUE-ds, 20M=20Minuten.

| Base model | Method | CS | FM | MB | PY | SQ | NC | ND | 20M | OP↑ | BWT↑ |
|---|---|---|---|---|---|---|---|---|---|---|---|
| | SeqLoRA | 42.3 | 45.0 | 23.0 | 40.3 | 66.7 | 54.3 | 50.8 | 42.9 | 45.7 | -7.2 |
| | LoRAMoE | 40.5 | 52.4 | 23.2 | 41.4 | 63.5 | 56.4 | 47.4 | 43.3 | 46.0 | -7.1 |
| | MH-MoE | **50.1** | **56.5** | **24.1** | 41.4 | **77.3** | **69.2** | 55.1 | **43.5** | **52.2** | **-2.2** |
| Llama-3.2-3B | MoLE | 34.7 | 25.0 | 21.4 | 29.2 | 67.1 | 38.3 | 40.9 | 43.3 | 37.5 | -11.5 |
| | LoRA-Mixer | 38.6 | 42.7 | 21.5 | 39.5 | 65.9 | 35.8 | 42.5 | **43.5** | 41.3 | -8.6 |
| | MixLoRA | 41.5 | 47.2 | 21.7 | **42.6** | 70.7 | 53.1 | 30.1 | 43.3 | 43.8 | -9.3 |
| | OPLoRA | 43.8 | 56.1 | 23.4 | 41.9 | 69.1 | 50.6 | **56.3** | 43.2 | 48.1 | -5.1 |
| | SeqLoRA | 44.6 | 60.3 | 18.3 | 44.8 | 66.4 | 56.7 | 69.5 | 43.0 | 50.5 | -8.2 |
| | LoRAMoE | 46.9 | 59.5 | 18.3 | 45.6 | 77.6 | 57.9 | 71.7 | 44.3 | 52.7 | -11.0 |
| | MH-MoE | **52.8** | **70.2** | **24.9** | **47.1** | **81.2** | 70.4 | **77.5** | **44.5** | **58.6** | **-1.7** |
| Ministral-3-8B | MoLE | 34.7 | 23.8 | 12.3 | 35.3 | 61.8 | 58.0 | 61.8 | 44.2 | 41.5 | -15.6 |
| | LoRA-Mixer | 34.8 | 26.2 | 13.6 | 37.4 | 56.1 | 56.8 | 61.2 | 44.1 | 41.3 | -16.0 |
| | MixLoRA | 48.9 | 53.6 | 13.0 | 44.5 | 77.7 | 70.4 | 62.5 | 42.7 | 51.7 | -8.5 |
| | OPLoRA | 51.5 | 65.3 | 23.6 | 45.9 | 74.1 | **72.8** | 74.8 | 42.3 | 56.3 | -3.9 |

**Benchmark.** We use TRACE (Wang et al., 2023), a continual-learning suite of eight diverse tasks. We follow the standard TRACE protocol and train sequentially over tasks.

**Base models.** We build MH-MoE on pretrained Llama-3.2-3B (Grattafiori et al., 2024) and Ministral-3-8B (Liu et al., 2026).

**Baselines.** We compare against (i) SeqLoRA, which sequentially trains a single shared LoRA adapter across tasks; (ii) LoRAMoE (Dou et al., 2024), a single-representation-routed MoE matched to MH-MoE in activated parameters per token; (iii) recent LoRA-MoE variants including MoLE (Huang et al., 2024), LoRA-Mixer (Li et al., 2025b) and MixLoRA (Li et al., 2024a); and (iv) continual-learning baseline OPLoRA (Xiong & Xie, 2026).

**Hardware.** Experiments are run on NVIDIA H100 GPUs and ZW810 PPUs.

**Metrics.** We report **Overall Performance (OP)**, the average score across all tasks after the final task, and **Backward Transfer (BWT)**, the mean change on earlier tasks after learning later ones. Higher OP and higher (less negative) BWT indicate better continual-learning performance.

Experimental details and additional results are provided in Appendix A.3 and A.4.

### 5.2. Main Results

**MH-MoE outperforms MoE-based routing variants.** Table 1 reports continual learning performance on TRACE after sequential training on all tasks. Compared with MoE-based routing variants, including LoRAMoE, MoLE, LoRA-Mixer, and MixLoRA, MH-MoE achieves the best Overall Performance (OP) and Backward Transfer (BWT) on

both backbones. On Llama-3.2-3B, MH-MoE improves OP from 46.0 to 52.2 over LoRAMoE and reduces forgetting from $-7.1$ to $-2.2$ BWT. On Ministral-3-8B, MH-MoE improves OP from 52.7 to 58.6 and BWT from $-11.0$ to $-1.7$. These results show that routing over post-attention sub-representations provides a more effective mechanism for mitigating interference than existing full-vector or adapter-composition MoE routing strategies.

**MH-MoE also improves over recent LoRA-based continual learning methods.** We further compare MH-MoE with SeqLoRA and OPLoRA, where OPLoRA is the strongest LoRA-based continual learning baseline in Table 1. On Llama-3.2-3B, OPLoRA achieves OP 48.1 and BWT $-5.1$, while MH-MoE further improves them to OP 52.2 and BWT $-2.2$. On Ministral-3-8B, OPLoRA achieves OP 56.3 and BWT $-3.9$, while MH-MoE improves them to OP 58.6 and BWT $-1.7$. This indicates that MH-MoE provides stronger retention than recent LoRA-based continual learning methods, while remaining task-agnostic and relying only on token-level routing.

### 5.3. Analysis

**MH-MoE reduces composition collision beyond route-space size.** We first examine whether the empirical gains of MH-MoE are explained by the mechanism identified in Section 2: reducing the collision of heterogeneous feature compositions within the same update destination. Using the same feature set as in Section 2.1, we compute the route/path-wise effective composition number $N_{\text{eff}}$. As shown in Fig. 6a, MH-MoE yields lower $N_{\text{eff}}$ than LoRAMoE, indicating that routing over post-attention sub-representations reduces composition mixing.

A possible alternative explanation is that MH-MoE reduces

mixing simply because tuple-valued routing creates a larger routing-outcome space. To rule this out, we construct a controlled comparison where the route-space cardinality and parameter count are kept comparable. Specifically, we compare MH-MoE with $M{=}8$ routing slices and per-slice top-1 routing among 4 head-private experts, yielding $4^8{=}65{,}536$ routing paths, against LoRAMoE with $K{=}26$ experts and top-5 routing, yielding $\binom{26}{5}{=}65{,}780$ possible expert sets. We also match the overall parameter budget between the two models. Under this setup, the dominant architectural difference is whether routing is performed *slice-wise on post-attention sub-representations* or *globally on the full head-mixed representation*. Table 2 shows that MH-MoE achieves better retention despite comparable route-space cardinality and capacity. Together with the lower $N_{\text{eff}}$ in Fig. 6a, this suggests that the gain comes from reducing composition collisions, rather than merely increasing the number of routing outcomes or model parameters.

We further consider a more constrained LoRAMoE baseline with $K{=}4$ experts and top-1 routing. Its routing-outcome space is much smaller, so more compositions are forced to share the same update destination. Consistently, the measured $N_{\text{eff}}$ increases in Fig. 6b, further supporting that $N_{\text{eff}}$ captures composition collision and tracks retention behavior across routing granularities.

**The gain does not come from arbitrary partitioning.** Since MH-MoE operates on contiguous post-attention slices, we next test whether its benefit comes from meaningful structure in these slices or merely from splitting the representation into smaller parts.

We introduce a permuted-slice control, where the representation dimensions are randomly permuted before slicing while keeping the same number of slices, experts, and routing structure. As shown in Table 3, permuted slicing performs much worse than MH-MoE on both Llama-2-7B-Instruct and Phi-4-mini-Instruct. On Llama-2-7B-Instruct, OP drops from 42.8 to 26.7 and BWT degrades from $-4.1$ to $-16.7$. On Phi-4-mini-Instruct (Abdin et al., 2024), OP drops from 48.4 to 26.8 and BWT degrades from $-6.7$ to $-26.7$.

These results indicate that MH-MoE does not benefit from arbitrary dimensional partitioning; rather, the structured post-attention slices used by MH-MoE are important for reducing harmful composition collisions.

**Head-specific routing and head-private capacity both contribute.** We further disentangle the effect of head-specific routing from that of private expert capacity. In the shared-expert-pool variant, each slice still uses its own router and routes based on its own sub-representation, but all slices select experts from a shared global expert pool instead of using head-private expert banks. Table 4 shows that this variant already improves substantially over stan-

dard LoRAMoE, indicating that routing on less mixed sub-representations is itself beneficial. However, full MH-MoE consistently performs better than the shared-pool variant. On Llama-2-7B-Instruct, the shared-pool variant improves OP/BWT from $37.5/-9.3$ to $41.2/-5.6$ over LoRAMoE, while full MH-MoE further improves them to $42.8/-4.1$. On Phi-4-mini-Instruct, the shared-pool variant improves OP/BWT from $41.9/-11.1$ to $46.7/-8.0$, while full MH-MoE reaches $48.4/-6.7$. Thus, routing on sub-representations drives much of the gain, while head-private expert banks further improve specialization by reducing cross-slice competition.

*Table 2.* **Routing-strategy ablation on TRACE using Qwen3-0.6B.**

| Method | CS | FM | MB | PY | SQ | NC | ND | 20M | OP↑ | BWT↑ |
|---|---|---|---|---|---|---|---|---|---|---|
| LoRAMoE | **49.2** | 40.5 | 15.3 | 48.8 | 69.4 | 30.9 | **52.6** | **37.8** | 43.1 | -9.3 |
| MH-MoE | 48.8 | **67.3** | **18.8** | **49.3** | **70.1** | **34.6** | 48.3 | 36.7 | **46.7** | **-4.5** |

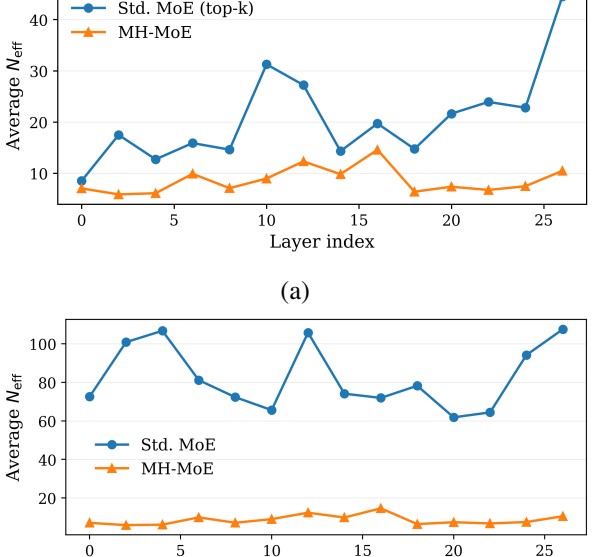

(a)

(b)

*Figure 6.* **Head-wise routing reduces composition collision.** (a) Under matched route-space size and activated budget (MH-MoE: $M{=}8$, top-1 over 4 head-private experts; LoRAMoE: $K{=}26$, top-5), MH-MoE yields lower route/path-wise mixing $N_{\text{eff}}$, indicating fewer semantic composition collisions. (b) With a much smaller route space (LoRAMoE: $K{=}4$, top-1), $N_{\text{eff}}$ increases, showing that constrained routing forces more compositions to share update destinations.

**MH-MoE mitigates forgetting consistently across task orderings.** Continual-learning performance can be sensitive to the task sequence, so we additionally test whether MH-MoE's retention gains persist under different TRACE orderings. Table 5 reports results for three task permutations. Across all orders, MH-MoE achieves higher OP and

*Table 3.* **Permuted-slice control.** Randomly permuting dimensions before slicing substantially degrades performance, showing that MH-MoE does not benefit from arbitrary partitioning.

| Backbone | Method | OP↑ | BWT↑ |
|---|---|---|---|
| Llama-2-7B-Instruct | MH-MoE | **42.8** | **-4.1** |
| | Perm-slice | 26.7 | -16.7 |
| Phi-4-mini-Instruct | MH-MoE | **48.4** | **-6.7** |
| | Perm-slice | 26.8 | -26.7 |

*Table 4.* **Shared-expert-pool ablation.** Head-specific routing already improves over LoRAMoE with a shared expert pool, while full MH-MoE further improves retention with head-private expert banks.

| Backbone | Method | OP↑ | BWT↑ |
|---|---|---|---|
| Llama-2-7B-Instruct | LoRAMoE | 37.5 | -9.3 |
| | MH-MoE (shared pool) | 41.2 | -5.6 |
| | MH-MoE | **42.8** | **-4.1** |
| Phi-4-mini-Instruct | LoRAMoE | 41.9 | -11.1 |
| | MH-MoE (shared pool) | 46.7 | -8.0 |
| | MH-MoE | **48.4** | **-6.7** |

substantially less negative BWT than LoRAMoE, indicating that the forgetting mitigation is not an artifact of a particular curriculum. This robustness is consistent with our mechanism: head-private routing reduces composition collision under diverse input streams, yielding more stable retention regardless of task order.

*Table 5.* **Task-ordering ablation on TRACE.**

| Order | Method | CS | FM | MB | PY | SQ | NC | ND | 20M | OP↑ | BWT↑ |
|---|---|---|---|---|---|---|---|---|---|---|---|
| 1 | LoRAMoE | 20.8 | 0.6 | 16.2 | 41.4 | 36.0 | 32.0 | 42.8 | 37.0 | 28.4 | -21.4 |
| | MH-MoE | **51.3** | **47.4** | **20.8** | **49.9** | **61.1** | **32.1** | **47.1** | 37.7 | **43.4** | **-8.0** |
| 2 | LoRAMoE | 34.1 | 0.2 | 17.0 | 46.4 | 49.3 | **21.0** | **30.2** | 37.2 | 29.4 | -18.3 |
| | MH-MoE | **49.2** | **55.6** | **20.6** | **50.0** | **57.4** | **21.0** | 26.5 | **37.6** | **39.7** | **-9.6** |
| 3 | LoRAMoE | 29.5 | 12.9 | 26.0 | 45.6 | 50.3 | 21.0 | 15.4 | **33.7** | 29.3 | -17.0 |
| | MH-MoE | **48.3** | **52.6** | **29.4** | **48.5** | **51.5** | **23.5** | **29.5** | 32.3 | **39.5** | **-10.9** |

**More heads increase routing granularity and improve overall performance.** We ablate the number of routing heads $M$ in MH-MoE. Increasing $M$ increases tuple-valued routing resolution (more virtual paths), which can reduce composition mixing, at the cost of additional routing/dispatch and reduced per-head capacity. Keeping other settings fixed, we evaluate $M \in \{2, 4, 8, 16\}$ on TRACE. Table 6 shows that OP improves with $M$ and is best at $M{=}16$.

*Table 6.* **MH-MoE head-count ablation on TRACE using Qwen3-8B.**

| $M$ | 2 | 4 | 8 | 16 |
|---|---|---|---|---|
| OP↑ | 53.1 | 52.6 | 54.5 | **56.9** |

*Table 7.* **End-to-end decoding efficiency.** We report decoding throughput, latency, peak memory, and achieved TFLOPS.

| Method | Tok/s↑ | ms/tok↓ | Mem (GB)↓ | TFLOPS↑ |
|---|---|---|---|---|
| LoRAMoE | 62.49 | 16.00 | 17.49 | 0.786 |
| MH-MoE ($M{=}8$) | 59.62 | 16.77 | 17.51 | 0.750 |
| MH-MoE ($M{=}16$) | 57.40 | 17.42 | 18.12 | 0.722 |

**Computation overhead.** Table 7 reports end-to-end decoding efficiency computed on ZW810 PPUs. Compared with LoRAMoE, MH-MoE with $M{=}8$ decreases throughput from 62.49 to 59.62 tok/s and increases latency from 16.00 to 16.77 ms/token, while peak memory remains nearly unchanged (17.49 GB vs. 17.51 GB). With $M{=}16$, the overhead increases moderately, yielding 57.40 tok/s, 17.42 ms/token, and 18.12 GB peak memory. These results indicate that MH-MoE trades a moderate amount of decoding efficiency for improved retention, with larger $M$ providing finer routing granularity at higher inference cost.

## 6. Limitations and Future Work

This work focuses on sequential continual learning on TRACE. Future studies should evaluate MH-MoE on longer task streams, broader benchmarks, and larger native MoE models. Theoretically, our analysis explains local composition collision and route-level forgetting susceptibility, but it does not fully characterize multi-step continual-learning dynamics. Extending the theory to long-horizon forgetting and accumulation effects is an important direction for future work.

## 7. Conclusion

We analyze catastrophic forgetting in MoE Transformers and show that routing on mixed head-structured signals causes composition collisions. We quantify this effect with the route-wise effective composition number $N_{\text{eff}}$ and link it to route-local forgetting. Based on this diagnosis, we propose MH-MoE, which performs head-wise routing to reduce composition mixing and improves retention on TRACE over LoRAMoE and other baselines.

## Impact Statement

This paper presents work whose goal is to advance the field of machine learning. There are many potential societal consequences of our work, none of which we feel must be specifically highlighted here.

## Acknowledgement

This research is supported in part by the National Natural Science Foundation of China under Grant 62090025.

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

# A. Appendix

### A.1. Head-wise Causal Importance

Fix a layer $\ell$. Let the router input at token $t$ be $h_t^{(\ell)} \in \mathbb{R}^d$, with $H$ heads and head dimension $d_h$ so $d = Hd_h$. We write

$$h_t^{(\ell)} = [h_{t,1}^{(\ell)}; \ldots; h_{t,H}^{(\ell)}], \qquad h_{t,m}^{(\ell)} \in \mathbb{R}^{d_h}.$$

For each feature $Y$, we train a linear probe $g_Y^{(\ell)} : \mathbb{R}^d \to \mathcal{Y}$ on $\{(h_i, y_i)\}_{i=1}^N$ and evaluate it with $\mathrm{Perf}(\cdot)$ (accuracy in our experiments).

**Head mean-replacement ablation.** Let $\mu_m^{(\ell)} \in \mathbb{R}^{d_h}$ be the empirical mean of head $m$'s block over the probe dataset:

$$\mu_m^{(\ell)} = \frac{1}{N} \sum_{i=1}^N h_{i,m}^{(\ell)}.$$

Define $\mathcal{A}_m : \mathbb{R}^d \to \mathbb{R}^d$ as replacing head $m$'s block by $\mu_m^{(\ell)}$:

$$\mathcal{A}_m(h) = [h_1; \ldots; h_{m-1}; \mu_m^{(\ell)}; h_{m+1}; \ldots; h_H].$$

**Importance score and shares.** We define head $m$'s causal importance for feature $Y$ at layer $\ell$ as the probe performance drop:

$$I_{Y,m}^{(\ell)} = \mathrm{Perf}\left(g_Y^{(\ell)}\right) - \mathrm{Perf}\left(g_Y^{(\ell)} \circ \mathcal{A}_m\right).$$

We normalize across heads to obtain a per-feature distribution over heads:

$$S_{Y,m}^{(\ell)} = \frac{I_{Y,m}^{(\ell)}}{\sum_{j=1}^H I_{Y,j}^{(\ell)} + \varepsilon}, \qquad \sum_{m=1}^H S_{Y,m}^{(\ell)} \approx 1,$$

with $\varepsilon$ a small constant for numerical stability.

### A.2. Proofs for Lemma 2.1 and Theorem 2.2

*Proof of Lemma 2.1.* Let $p(\cdot \mid r)$ be the distribution on $\mathcal{C}$, and let $S \subseteq \mathcal{C}$ with $|S| \leq m$. By Cauchy–Schwarz,

$$\sum_{c \in S} p(c \mid r) = \langle \mathbf{1}_S, \, p(\cdot \mid r) \rangle \leq \|\mathbf{1}_S\|_2 \, \|p(\cdot \mid r)\|_2 = \sqrt{|S|} \sqrt{\sum_{c \in \mathcal{C}} p(c \mid r)^2} \leq \sqrt{m} \sqrt{\sum_{c \in \mathcal{C}} p(c \mid r)^2}.$$

Using $N_{\mathrm{eff}}(r) = \left(\sum_{c \in \mathcal{C}} p(c \mid r)^2\right)^{-1}$, we obtain

$$\Pr_{C \sim p(\cdot|r)}[C \in S] = \sum_{c \in S} p(c \mid r) \leq \sqrt{\frac{m}{N_{\mathrm{eff}}(r)}}.$$

Therefore,

$$\Pr_{C \sim p(\cdot|r)}[C \notin S] = 1 - \Pr_{C \sim p(\cdot|r)}[C \in S] \geq 1 - \sqrt{\frac{m}{N_{\mathrm{eff}}(r)}}.$$

$\square$

*Proof of Theorem 2.2.* Fix route $r$, step size $\eta > 0$, and a (possibly random) unit update direction $\hat{u}$ with $\theta^+ = \theta - \eta\hat{u}$. For each $c \in \mathcal{C}$, define the one-step change $\Delta_c := F_c(\theta^+) - F_c(\theta)$.

**Step 1: a uniform lower bound from smoothness and bounded gradients.** Since each $F_c$ is $L$-smooth, for any vector $v$ we have the quadratic lower bound

$$F_c(\theta + v) \geq F_c(\theta) + \langle \nabla F_c(\theta), v \rangle - \frac{L}{2}\|v\|_2^2.$$

Applying this with $v = -\eta\hat{u}$ yields

$$F_c(\theta - \eta\hat{u}) \geq F_c(\theta) - \eta\langle \nabla F_c(\theta), \hat{u} \rangle - \frac{L\eta^2}{2}.$$

Thus,

$$\Delta_c \geq -\eta\langle \nabla F_c(\theta), \hat{u} \rangle - \frac{L\eta^2}{2}.$$

Using $\|\hat{u}\| = 1$ and the assumption $\|\nabla F_c(\theta)\| \leq G$,

$$-\langle \nabla F_c(\theta), \hat{u} \rangle \geq -\|\nabla F_c(\theta)\|\,\|\hat{u}\| \geq -G,$$

hence for every $c \in \mathcal{C}$,

$$\Delta_c \geq -\eta G - \frac{L\eta^2}{2}. \tag{18}$$

**Step 2: expected per-composition lower bound for $c \notin S$.** Let $c \notin S$. By assumption, $\Pr(\Delta_c \geq \kappa) \geq \rho$, where the probability is over the randomness defining $\hat{u}$. Decomposing by events and using (18),

$$\mathbb{E}[\Delta_c] = \mathbb{E}[\Delta_c \mathbf{1}\{\Delta_c \geq \kappa\}] + \mathbb{E}[\Delta_c \mathbf{1}\{\Delta_c < \kappa\}] \geq \kappa\,\Pr(\Delta_c \geq \kappa) + \left(-\eta G - \frac{L\eta^2}{2}\right)\Pr(\Delta_c < \kappa).$$

Therefore,

$$\mathbb{E}[\Delta_c] \geq \rho\kappa - (1-\rho)\left(\eta G + \frac{L\eta^2}{2}\right), \qquad \forall c \notin S. \tag{19}$$

**Step 3: lift to the route mixture using Lemma 2.1.** By definition,

$$F_r(\theta) = \mathbb{E}_{C \sim p(\cdot|r)}[F_C(\theta)] \quad \Rightarrow \quad F_r(\theta^+) - F_r(\theta) = \mathbb{E}_{C \sim p(\cdot|r)}[\Delta_C].$$

Taking expectation over the randomness defining $\hat{u}$ and using iterated expectation,

$$\mathbb{E}[F_r(\theta^+) - F_r(\theta)] = \sum_{c \in \mathcal{C}} p(c \mid r)\,\mathbb{E}[\Delta_c].$$

Split the sum into $S$ and $\mathcal{C} \setminus S$:

$$\mathbb{E}[F_r(\theta^+) - F_r(\theta)] = \sum_{c \notin S} p(c \mid r)\,\mathbb{E}[\Delta_c] + \sum_{c \in S} p(c \mid r)\,\mathbb{E}[\Delta_c].$$

For $c \notin S$, Eq. (19) gives

$$\mathbb{E}[\Delta_c] \geq B_{\text{out}} := \rho\kappa - (1-\rho)\left(\eta G + \frac{L\eta^2}{2}\right).$$

For $c \in S$, we cannot assume $\mathbb{E}[\Delta_c] \geq 0$, but the uniform bound Eq. (18) yields

$$\mathbb{E}[\Delta_c] \geq -B_{\text{in}} := -\left(\eta G + \frac{L\eta^2}{2}\right).$$

Therefore,

$$\mathbb{E}[F_r(\theta^+) - F_r(\theta)] \geq \Pr[C \notin S] \cdot B_{\text{out}} - \Pr[C \in S] \cdot B_{\text{in}}.$$

Let $a := \sqrt{m/N_{\text{eff}}(r)}$. By Lemma 2.1, $\Pr[C \in S] \leq a$ and $\Pr[C \notin S] \geq (1-a)_+$, hence

$$\mathbb{E}[F_r(\theta^+) - F_r(\theta)] \geq (1-a)_+ B_{\text{out}} - a B_{\text{in}}.$$

**Monotonicity in $N_{\text{eff}}(r)$ and positivity.** Let $a = \sqrt{m/N_{\text{eff}}(r)}$, so $a$ decreases as $N_{\text{eff}}(r)$ increases. Our bound can be written as

$$\text{LB}(a) = (1 - a)_+ B_{\text{out}} - a B_{\text{in}},$$

where $B_{\text{out}} = \rho\kappa - (1 - \rho)\big(\eta G + \frac{L\eta^2}{2}\big)$ and $B_{\text{in}} = \eta G + \frac{L\eta^2}{2}$. When $a < 1$, $\text{LB}(a) = B_{\text{out}} - a(B_{\text{out}} + B_{\text{in}})$, which is nondecreasing in $N_{\text{eff}}(r)$ (equivalently, nonincreasing in $a$) whenever $B_{\text{out}} + B_{\text{in}} > 0$. Moreover, if $B_{\text{out}} > 0$, the bound becomes positive once

$$a < \frac{B_{\text{out}}}{B_{\text{out}} + B_{\text{in}}} \quad \Longleftrightarrow \quad N_{\text{eff}}(r) > m\Big(\frac{B_{\text{out}} + B_{\text{in}}}{B_{\text{out}}}\Big)^2.$$

$\square$

### A.3. Experiment Details

#### A.3.1. DATASETS

**C-STANCE.** C-STANCE is a large-scale Chinese benchmark for zero-shot stance detection, where each example pairs a microblog post with a target and the model predicts a stance label (favor/against/neutral) toward that target, including targets not observed during training. In TRACE, C-STANCE is cast as a 3-way classification task and is evaluated using accuracy.

**FOMC.** The FOMC dataset is constructed from Federal Open Market Committee communications and is annotated for monetary-policy stance, enabling a hawkish–dovish style classification task. TRACE uses this dataset as a domain-specific stance classification problem (English) and reports accuracy.

**MeetingBank.** MeetingBank is a meeting summarization dataset built from city council meetings, providing long-form transcripts together with professionally written minutes and aligned segment-level supervision via a divide-and-conquer alignment procedure. TRACE uses MeetingBank as an abstractive summarization task and evaluates generation quality with ROUGE-L.

**Py150.** Py150 is a corpus of 150,000 Python source files mined from GitHub under permissive licensing filters and quality controls. It is widely used as a standard benchmark for Python code completion. TRACE uses Py150 as a code generation/completion task and evaluates with the fuzzing accuracy.

**ScienceQA.** ScienceQA is a science question answering benchmark containing multiple-choice questions collected from school science curricula. TRACE uses ScienceQA as a discrete-answer QA task and reports accuracy.

**NumGLUE-cm.** NumGLUE is a suite of arithmetic-centric reasoning tasks. TRACE includes the NumGLUE *Commonsense + Arithmetic* task (cm), which requires combining commonsense quantitative facts with simple arithmetic operations. TRACE evaluates this task using accuracy.

**NumGLUE-ds.** TRACE also includes the NumGLUE *Domain Specific + Arithmetic* task (ds), which requires domain knowledge together with arithmetic reasoning. As with other discrete-answer tasks in TRACE, performance is reported using accuracy.

**20Minuten.** 20Minuten is a German dataset collected from the Swiss news outlet *20 Minuten*, pairing full news articles with simplified rewrites/summaries to support document-level text simplification. TRACE uses 20Minuten as a German generation task and evaluates outputs using SARI.

#### A.3.2. BASELINES

**SeqLoRA.** SeqLoRA is an adapter-based continual learning baseline that equips the pretrained model with a *single shared* set of LoRA adapters. The same LoRA parameters are trained sequentially across all tasks in the stream, while the pretrained backbone remains frozen.

**LoRAMoE (Dou et al., 2024).** LoRAMoE replaces selected linear layers with a shared *common* linear map plus $K$ LoRA-style low-rank residual experts and a learned gate. Routing is computed from a single router input representation, and each token activates a sparse subset of experts; the output is the common branch plus the probability-weighted residual from the selected experts. During continual learning, we freeze the pretrained backbone and train only the residual experts and gate parameters. We match LoRAMoE to MH-MoE in activated parameter count per token.

**MoLE (Huang et al., 2024).** MoLE composes multiple trained LoRA modules by treating the LoRA module at each layer as an expert and learning a layer-wise gating function to determine the composition weights. Unlike linear arithmetic LoRA composition, which uses fixed weights and may dilute the characteristics of individual LoRAs, MoLE learns hierarchical, layer-specific mixture weights while keeping the pretrained backbone and LoRA experts frozen. In our continual learning setting, we adapt MoLE by first training task-specific LoRA experts and then training only the layer-wise gates during sequential learning.

**MixLoRA (Li et al., 2024a).** MixLoRA constructs a parameter-efficient sparse MoE model by inserting multiple LoRA-based experts into the feed-forward network blocks of a frozen pretrained model. It uses a top-$k$ router to select LoRA experts for each token and combines the selected expert outputs with the shared FFN branch. MixLoRA also introduces attention-layer LoRA adapters and an auxiliary load-balancing loss to reduce routing imbalance. In our implementation, we follow its LoRA-MoE design and train the router and LoRA experts under the same continual learning protocol as the other adapter-based baselines.

**LoRA-Mixer (Li et al., 2025b).** LoRA-Mixer is a modular LoRA-MoE method that applies dynamically routed LoRA experts to the linear projection layers of the attention module, rather than replacing whole attention/FFN layers or adding only parallel expert branches. It supports both jointly trained LoRA experts and reusable pretrained LoRA modules, and employs a hard-soft routing strategy together with a specialization-balance loss to encourage task-aware expert usage. For continual learning, we adapt LoRA-Mixer by using task-specific LoRA experts and updating the routing module sequentially while keeping the pretrained backbone frozen.

**OPLoRA (Xiong & Xie, 2026).** OPLoRA is a LoRA-based continual learning baseline that aims to mitigate catastrophic forgetting by preventing LoRA updates from interfering with the dominant singular subspaces of the pretrained weights. For each frozen weight matrix, OPLoRA performs singular value decomposition and constructs double-sided orthogonal projectors from the top-$k$ left and right singular vectors. The LoRA update is then constrained as $\Delta W = P_L B A P_R$, where $P_L = I - U_k U_k^\top$ and $P_R = I - V_k V_k^\top$. This projection forces the update to lie in the orthogonal complement of the dominant pretrained subspace, thereby preserving the top-$k$ singular triples while allowing task-specific adaptation in the residual subspace. In our implementation, we freeze the pretrained backbone and train only the projected LoRA parameters under the same sequential continual learning protocol as other adapter-based baselines.

### A.3.3. METRICS

Let $f_i(\mathbf{w}_j)$ denote the prediction performance on task $i$ (e.g., accuracy, SARI, ROUGE-L) when evaluated using the model parameters after learning through task $j$, denoted by $\mathbf{w}_j$.

**Overall Performance (OP).** After training up to task $n$, we define the overall performance as the average score over all tasks seen so far:

$$\text{OP}_n \triangleq \frac{1}{n} \sum_{i=1}^{n} f_i(\mathbf{w}_n). \tag{20}$$

$\text{OP}_n$ measures how good the final model $\mathbf{w}_n$ is on the full set of tasks $\{1, \ldots, n\}$. Higher $\text{OP}_n$ means better overall learning quality.

**Backward Transfer (BWT).** We quantify forgetting via backward transfer, defined as the average performance drop on earlier tasks after learning all tasks:

$$\text{BWT}_n \triangleq \frac{1}{n} \sum_{i=1}^{n} \Big( f_i(\mathbf{w}_n) - f_i(\mathbf{w}_i) \Big). \tag{21}$$

For each task $i$, the term $f_i(\mathbf{w}_i)$ is the model's performance right after learning task $i$, while $f_i(\mathbf{w}_n)$ is its performance on task $i$ after subsequently learning tasks $i+1, \ldots, n$. Thus, $\text{BWT}_n$ measures *retention*: Negative values indicate forgetting, values closer to zero indicate better retention, and higher BWT is better.

### A.3.4. IMPLEMENTATION DETAILS

We train each task for 5 epochs with a learning rate of $1 \times 10^{-4}$ under a cosine learning-rate schedule. In all experiments, we fix the sequence length to 2048 and use a batch size of 10. We optimize with AdamW (`weight_decay = 0.01`, $\beta_1 = 0.9$, $\beta_2 = 0.95$, $\epsilon = 10^{-6}$). We attach experts to the linear modules in the MLP layers and tune the LoRA rank so that the number of activated parameters is comparable across methods. For LoRAMoE and MixLoRA, we use 4 experts per layer. For MH-MoE, each head has 4 private experts and performs top-1 routing within its head-specific expert set. For MoLE and LoRA-Mixer, we first train LoRA modules for each task individually, then train the gating modules in a sequential continual learning setup. For OPLoRA, we set the regularization weight to 0.05.

Unless stated otherwise, we use the standard TRACE task order: C-STANCE $\rightarrow$ FOMC $\rightarrow$ MeetingBank $\rightarrow$ Py150 $\rightarrow$ ScienceQA $\rightarrow$ NumGLUE-cm $\rightarrow$ NumGLUE-ds $\rightarrow$ 20Minuten. For the task-order ablation, we evaluate three alternative sequences: **Order 1**: FOMC $\rightarrow$ C-STANCE $\rightarrow$ ScienceQA $\rightarrow$ Py150 $\rightarrow$ MeetingBank $\rightarrow$ NumGLUE-cm $\rightarrow$ NumGLUE-ds $\rightarrow$ 20Minuten; **Order 2**: FOMC $\rightarrow$ C-STANCE $\rightarrow$ ScienceQA $\rightarrow$ Py150 $\rightarrow$ NumGLUE-ds $\rightarrow$ NumGLUE-cm $\rightarrow$ MeetingBank $\rightarrow$ 20Minuten; **Order 3**: FOMC $\rightarrow$ C-STANCE $\rightarrow$ ScienceQA $\rightarrow$ Py150 $\rightarrow$ MeetingBank $\rightarrow$ NumGLUE-ds $\rightarrow$ NumGLUE-cm $\rightarrow$ 20Minuten.

## A.4. Additional Experiment Results

### A.4.1. HEAD-STRUCTURED FEATURES. (SECTION 2.1)

Figure 7 extends the head-wise causal-importance analysis in Section 2.1 to multiple layers. Across layers, feature importance is consistently non-uniform over heads: each feature is mainly supported by a small subset of heads, and the dominant heads differ across features. This supports the claim that the post-attention router input is not only multi-feature, but also head-structured, motivating routing over sub-representations rather than using a single full-vector routing decision.

### A.4.2. WITHIN-COMPOSITION COHERENCE VS. CROSS-COMPOSITION WEAK ALIGNMENT. (SECTION 2.2)

Figure 8 extends the gradient-direction analysis in Section 2.2 across layers. Within-composition splits show consistently higher cosine similarity than cross-composition pairs, while cross-composition similarities concentrate closer to zero. This indicates that tokens sharing the same feature composition induce more coherent update directions, whereas different compositions tend to produce weakly aligned learning signals. Therefore, routing multiple heterogeneous compositions to the same expert can create gradient conflict and increase forgetting risk.

### A.4.3. EXPANDED ANALYSIS OF WITHIN-COMPOSITION COHERENCE VS. CROSS-COMPOSITION WEAK ALIGNMENT ON OTHER TRACE TASKS. (SECTION 2.2)

To further examine whether the composition-separation phenomenon observed in Section 2.2 is specific to the two analyzed tasks or holds more broadly, we extend the gradient-alignment analysis to all eight TRACE tasks. The results are shown in Table 8. Across all tasks, the within-composition gradient alignment is consistently higher than the across-composition gradient alignment. This indicates that gradients induced by samples sharing the same composition are more coherent than gradients induced by samples from different compositions, supporting the generality of the within-composition coherence and cross-composition weak-alignment pattern.

At the same time, the strength of this phenomenon is task-dependent. The within-minus-across gap is larger on ScienceQA, NumGLUE-ds, NumGLUE-cm, FOMC, and C-STANCE, suggesting stronger composition-level gradient separation on these tasks. In contrast, the gap is smaller on Py150, MeetingBank, and 20Minuten, where the empirical performance difference between MH-MoE and LoRAMoE is also less pronounced. This is consistent with our interpretation that MH-MoE is most beneficial when the task exhibits stronger composition-dependent interference under full-vector routing.

We also compute the effective composition number $N_{\text{eff}}$ for both LoRAMoE and MH-MoE on all eight TRACE tasks. As shown in Table 9, MH-MoE consistently yields a much smaller $N_{\text{eff}}$ than LoRAMoE across the entire benchmark. This suggests that head-wise routing substantially reduces the number of distinct compositions mixed within the same route, thereby alleviating the composition-collision effect induced by full-vector routing.

*Table 8.* Gradient alignment across all eight TRACE tasks. "Within" denotes the average gradient alignment between samples from the same composition, "Across" denotes the average gradient alignment between samples from different compositions, and "Gap" denotes their difference.

| Task | Within | Across | Gap |
|------|--------|--------|-----|
| ScienceQA | 0.683 | 0.026 | 0.657 |
| Py150 | 0.288 | 0.021 | 0.268 |
| NumGLUE-ds | 0.907 | 0.076 | 0.831 |
| NumGLUE-cm | 0.754 | 0.070 | 0.684 |
| MeetingBank | 0.300 | 0.016 | 0.284 |
| FOMC | 0.741 | 0.057 | 0.684 |
| C-STANCE | 0.606 | 0.059 | 0.546 |
| 20Minuten | 0.364 | 0.028 | 0.336 |

*Table 9.* Comparison of effective composition number $N_{\text{eff}}$ across all eight TRACE tasks. Lower values indicate that fewer compositions are mixed within the same route.

| Task | LoRAMoE | MH-MoE |
|------|---------|--------|
| ScienceQA | 269.09 | 10.74 |
| Py150 | 132.49 | 9.50 |
| NumGLUE-ds | 103.41 | 3.77 |
| NumGLUE-cm | 64.90 | 3.35 |
| MeetingBank | 270.42 | 15.15 |
| FOMC | 72.04 | 5.71 |
| C-STANCE | 93.96 | 3.99 |
| 20Minuten | 238.59 | 5.00 |

Overall, these expanded results show that the proposed diagnosis is not limited to C-STANCE and FOMC. Across TRACE, samples from the same composition consistently exhibit stronger gradient coherence than samples from different compositions, and MH-MoE consistently reduces route-level composition mixing compared with LoRAMoE. The task-dependent variation in the alignment gap further explains why the advantage of MH-MoE is more pronounced on some tasks than on others.

### A.4.4. EXPERIMENT RESULTS ON OTHER BACKBONE MODELS.

*Table 10.* **Continual learning performance on TRACE after training on all tasks for additional backbones.** We report the final score on each dataset, Overall Performance (OP), and Backward Transfer (BWT). Abbreviations: CS=C-STANCE, FM=FOMC, MB=MeetingBank, PY=Py150, SQ=ScienceQA, NC=NumGLUE-cm, ND=NumGLUE-ds, 20M=20Minuten.

| Base model | Method | CS | FM | MB | PY | SQ | NC | ND | 20M | OP↑ | BWT↑ |
|------------|--------|----|----|----|----|----|----|----|-----|-----|------|
| Qwen3-0.6B | SeqLoRA | 12.8 | 26.2 | 14.6 | 43.5 | 59.8 | 7.4 | 34.8 | 36.6 | 29.5 | -19.0 |
|  | LoRAMoE | 34.3 | 52.0 | 12.0 | 46.3 | 61.4 | 14.8 | 44.3 | 37.2 | 37.8 | -11.2 |
|  | MH-MoE | 48.8 | 67.3 | 18.8 | 49.3 | 70.1 | 34.6 | 48.3 | 36.7 | 46.7 | -4.5 |
| Qwen3-8B | SeqLoRA | 49.6 | 56.7 | 16.7 | 49.7 | 88.8 | 66.7 | 63.7 | 40.2 | 54.0 | -7.3 |
|  | LoRAMoE | 49.1 | 65.7 | 14.6 | 52.8 | 89.2 | 71.6 | 58.8 | 39.2 | 55.1 | -5.5 |
|  | MH-MoE | 50.5 | 65.9 | 18.2 | 50.9 | 90.8 | 67.9 | 70.5 | 40.7 | 56.9 | -5.1 |
| Llama-2-7B-Instruct | SeqLoRA | 24.2 | 50.2 | 18.7 | 31.6 | 60.5 | 34.6 | 26.5 | 43.2 | 36.2 | -9.7 |
|  | LoRAMoE | 24.7 | 54.8 | 20.4 | 31.8 | 60.8 | 34.6 | 28.6 | 44.0 | 37.5 | -9.3 |
|  | MH-MoE | 46.2 | 61.9 | 21.6 | 32.3 | 64.4 | 38.3 | 33.8 | 43.6 | 42.8 | -4.1 |
| Phi-4-mini-Instruct | SeqLoRA | 38.6 | 54.6 | 7.2 | 31.8 | 64.9 | 56.8 | 32.3 | 43.0 | 41.2 | -11.6 |
|  | LoRAMoE | 38.2 | 55.6 | 9.3 | 32.1 | 65.8 | 60.4 | 31.1 | 42.6 | 41.9 | -11.1 |
|  | MH-MoE | 49.3 | 68.3 | 9.5 | 32.9 | 77.9 | 65.4 | 40.6 | 43.2 | 48.4 | -6.7 |

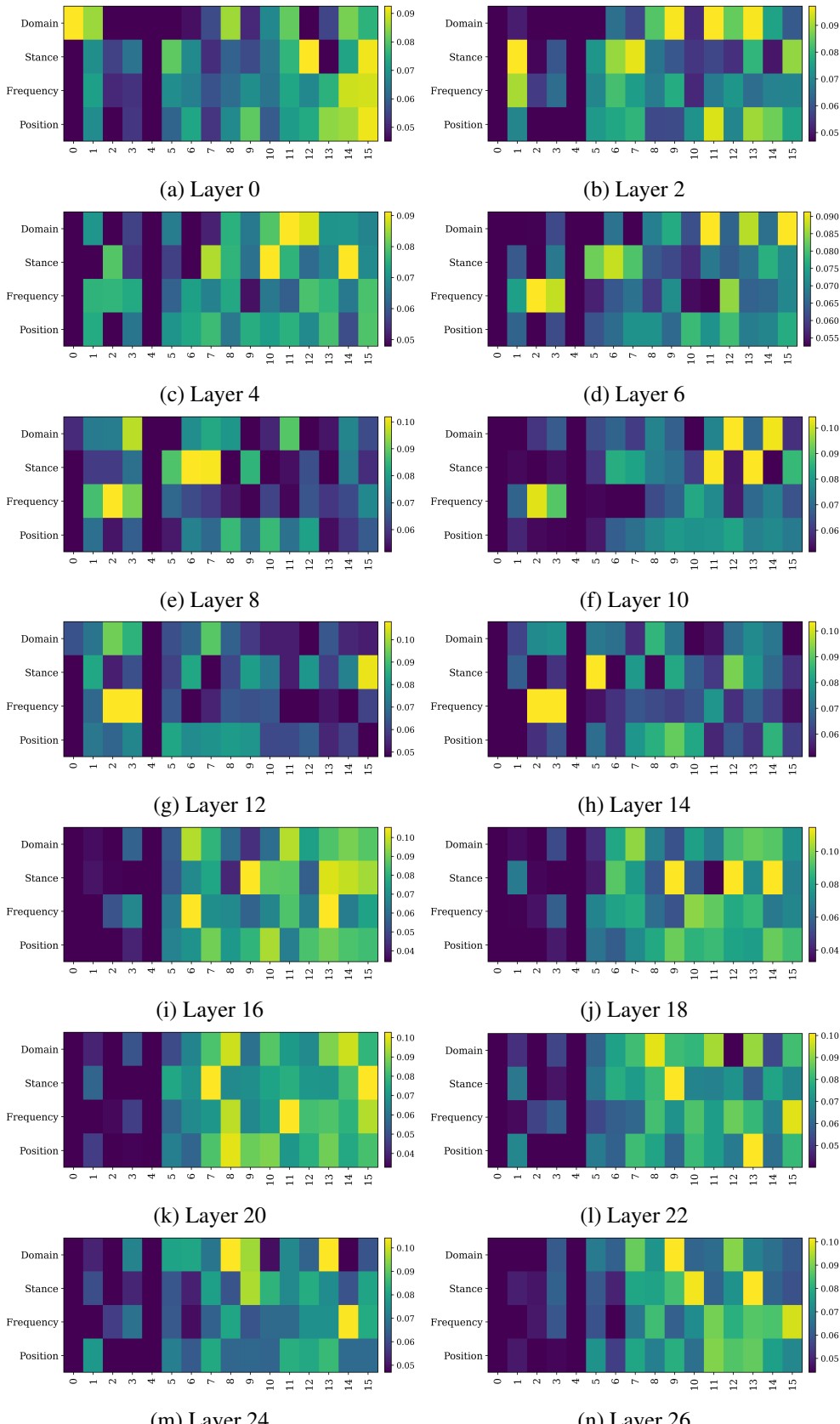

*Figure 7.* **Feature signals are head-structured across model layers.**

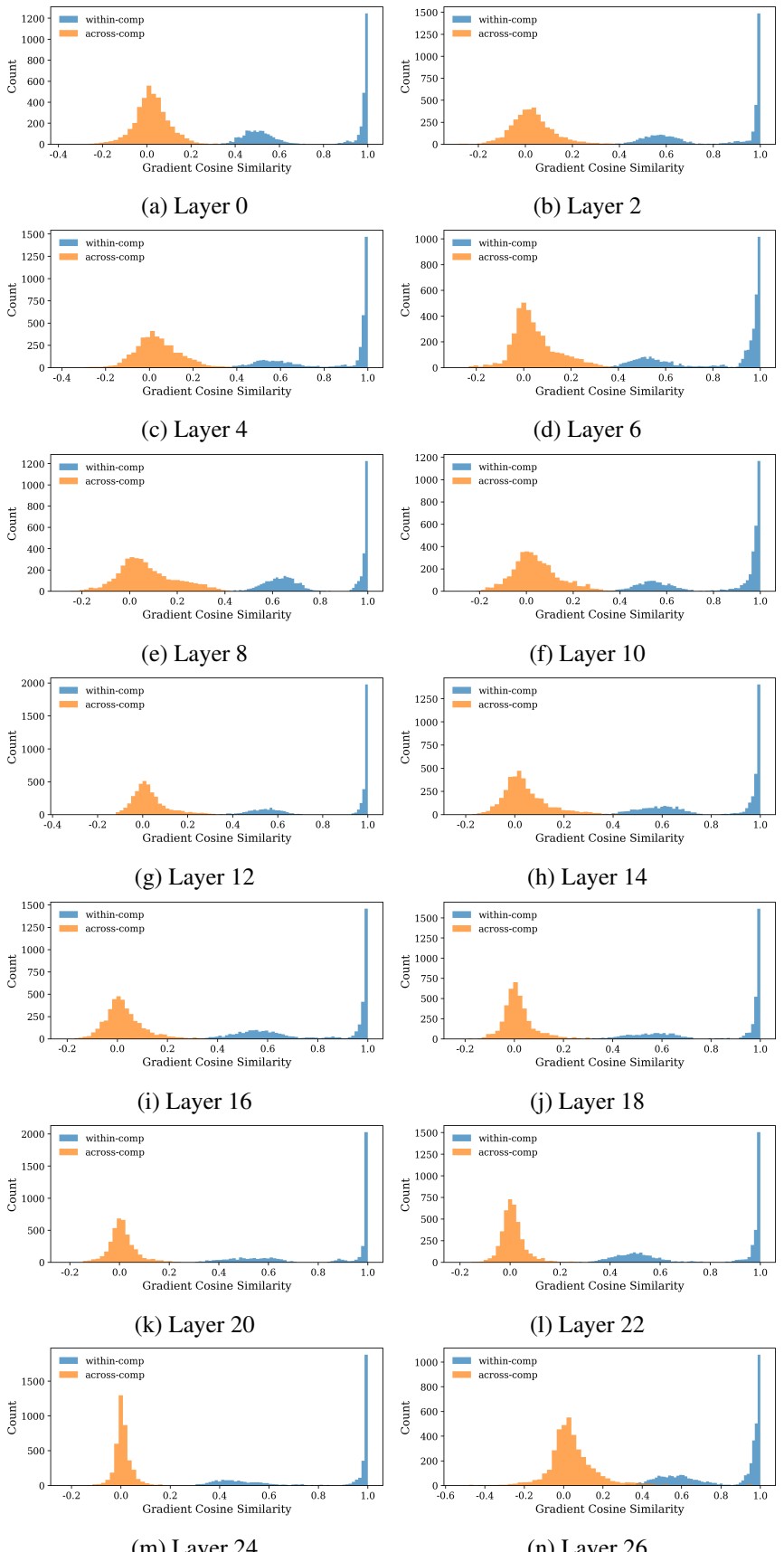

*Figure 8.* **Different feature compositions induce distinct gradient directions across layers.**

