# OpenReview forum: "Multi-Head Attention as a Source of Catastrophic Forgetting in MoE Transformers"
_ICML.cc/2026/Conference — ICML 2026 regular_

### Official Review · Reviewer_mn6b · 2026-02-26

**Soundness:** 2
**Presentation:** 2
**Significance:** 2
**Originality:** 1
**Overall Recommendation:** 4
**Confidence:** 5

**Summary:**

This paper studies catastrophic forgetting in MoE Transformers under continual learning. It argues that multi-head attention mixes head-specific signals before routing, causing feature-composition collisions that increase interference. To address this, the authors propose MH-MoE, which performs head-wise routing to reduce such collisions. Experiments on TRACE show improved retention over LoRAMoE, including BWT on Qwen3-0.6B improving from -11.2% to -4.5%.

**Compliance With Llm Reviewing Policy:**

Affirmed.

**Final Justification:**

Thank you for the author's reply and additional experiments. My problem has been solved. I hope the author can add the above experiments to the main text in the final draft. These experiments can effectively enhance the quality of the paper. Regarding the innovativeness, I still have doubts. I hope the author can show the difference from other methods in the main text later, for example, by adding a table to highlight their innovativeness. I will revise my score to 4.

**Key Questions For Authors:**

Please refer weaknesses.

**Limitations:**

Yes

**Strengths And Weaknesses:**

Strength:

The method is generally simple and easy to follow. The motivation is reasonable.

Weaknesses:

1: I feel this paper's contribution is quite limited. Essentially, it only addresses routing at the head level rather than proposing a new paradigm or framework, thus its innovation is rather limited.

2: In terms of writing style, I believe this paper doesn't need excessive theoretical proof; I think it's over-packaging and counterproductive.

3: In experimental validation, the baselines used by the authors are outdated. Baselines from 2024-2026 are not included, and some strong LoRA-MoE baselines such as MoLE[1], LoRA-Mixer[2], and MixLoRA[3] are not cited or incorporated.

4: Regarding model selection, the authors only used Qwen series MoE models. Other series of MoE models should also be included to demonstrate their generality.

[1] https://arxiv.org/abs/2404.13628

[2] https://arxiv.org/abs/2507.00029

[3] https://arxiv.org/abs/2404.15159

---

> ### Author Rebuttal · Authors · 2026-03-30
>
> We thank the reviewer for the careful reading. We address the concerns on contribution scope, role of the theory, baseline selection, and generality below.
>
> **W1 (Contribution scope):** Our contribution is twofold: (i) a concrete mechanism of forgetting in continual learning with MoE Transformers is identified and validated—routing on post-attention representations mixes multiple semantic and structural factors before expert assignment, increasing composition collisions and interference; and (ii) MH-MoE is proposed as a direct architectural response to this mechanism by performing head-wise routing on sub-representations. In this sense, the paper is not only a routing modification, but a diagnosis-to-design contribution: it explains a concrete source of forgetting in standard MoE Transformers and introduces a matching remedy that improves retention across multiple backbones. We will revise the manuscript to state this contribution more explicitly.
>
> **W2 (Role of the theory):** The theoretical part is included to formalize the paper’s central mechanism. Its role is to show that the connection between route-level composition mixing and forgetting is not only an empirical correlation, but also has a clear analytical basis. In particular, Lemma 2.1 shows that when the effective composition number $N_{\mathrm{eff}}(r)$ is large, the old-task mass on route $r$ cannot be concentrated on a small protected subset of compositions. Theorem 2.2 then shows that, under standard smoothness and weak cross-composition alignment assumptions, larger $N_{\mathrm{eff}}(r)$ leads to a stronger lower bound on the expected old-task loss increase on that route after an update. We will revise the manuscript to make this motivation clearer.
>
> **W3 (Baseline selection):** We will explicitly discuss these works in the revision. Notably, our experimental setting is sequential continual learning with explicit forgetting evaluation, including BWT, which is fundamentally different from the targeted settings of these methods. MoLE focuses on composing trained LoRA modules, MixLoRA on efficient sparse LoRA-MoE fine-tuning, and LoRA-Mixer on modular composition and transfer of pretrained LoRA modules. After a careful literature survey, the baselines in the current submission were chosen as the closest and most appropriate methods we could identify for the sequential-forgetting setting studied here. We will revise the manuscript to make this distinction explicit. If the reviewer is aware of additional baselines that are closer to this setting, we would welcome further discussion.
>
> **W4 (Generality across backbones):** To validate robustness and generality, additional experiments were conducted on Phi-4-mini-Instruct and Llama-2-7B-Instruct. In both cases, MH-MoE again outperforms LoRAMoE, indicating that the benefit is not tied to a single backbone family. We will add these results and revise the manuscript accordingly.
>
> |Model|Method|CS|FM|MB|PY|SQ|NC|ND|20M|OP|BWT|
> |---|---|---:|---:|---:|---:|---:|---:|---:|---:|---:|---:|
> |Llama-2-7B-Instruct|SeqLoRA|24.2|50.2|18.7|31.6|60.5|34.6|26.5|43.2|36.2|-9.7|
> ||LoRAMoE|24.7|54.8|20.4|31.8|60.8|34.6|28.6|44.0|37.5|-9.3|
> ||MH-MoE|46.2|61.9|21.6|32.3|64.4|38.3|33.8|43.6|42.8|-4.1|
> |Phi-4-mini-Instruct|SeqLoRA|38.6|54.6|7.2|31.8|64.9|56.8|32.3|43.0|41.2|-11.6|
> ||LoRAMoE|38.2|55.6|9.3|32.1|65.8|60.4|31.1|42.6|41.9|-11.1|
> ||MH-MoE|49.3|68.3|9.5|32.9|77.9|65.4|40.6|43.2|48.4|-6.7|

---

> > ### Author Rebuttal · Reviewer_mn6b · 2026-04-01
> >
> > 1: Regarding innovation, I agree with the authors' claim of "from diagnosis to design," and the design is reasonable and targeted. However, I believe that, in terms of the methodology itself, this paper's contribution remains incremental. MH-MoE is still a slight modification of existing LoRA-MoE and MoE-routing methods. Specifically, MH-MoE changes single full-vector routing to head-private subrepresentation routing, alters the granularity of expert allocation, but doesn't introduce any new CL paradigm or change the basic structure of MoE.
> >
> > 2: Regarding the choice of baseline, I disagree with the authors' response. The combination of LoRA and MoE is itself a method for addressing continuous model learning (forgetting resistance). Comparing this paper only to LoRAMoE is unacceptable. The baselines I proposed simply didn't explore the forgetting problem, but their methods themselves can resist forgetting. Since the authors are comparing with LoRAMoE, the entire LoRA+MoE series of methods should be widely included in the comparison. Moreover, the forgetting resistance methods the authors compared, such as O-LoRA and EWC, are very old baselines, which are insufficient for current applications.
> >
> > 3: I appreciate the authors' additional experiments on Llama and Phi, but I don't understand why they continued experimenting on such an outdated model. Experiments on newer, mainstream models like Qwen3, Llama3.2, and Mistral are far more convincing.
> >
> > 4: Regarding the theoretical proof, I appreciate the author's clarification. Having carefully reviewed the theory, I have the following concerns: Lemma 2.1 seems to be merely a basic mixed-quality bound, insufficient on its own to establish a strong causal relationship between routing conflicts and forgetting. More importantly, Theorem 2.2 relies on a strong assumption: for combinations outside a small, well-aligned subset, the updated loss of the old task increases by at least $\kappa$ with a probability of at least $\rho$. In my view, this assumption already encompasses most of the expected outcome that combinations with fewer protections will be. Therefore, the theorem is more like a statement of conditional susceptibility. While intuitive, it doesn't derive the forgetting mechanism from the model itself, so I'm unsure if these forgotten theories are strongly relevant to the current paper. Furthermore, the theoretical analysis is based on single-step updates, while catastrophic forgetting in continuous learning is a multi-step cumulative phenomenon. For these reasons, I still believe the theoretical portion is somewhat overemphasized relative to the proposed method.
> >
> > I would increase the score if the author could address these issues.

---

> > > ### Author Response · Authors · 2026-04-04
> > >
> > > Each point is addressed below.
> > >
> > > **A1:** Thank you for the reviewer's recognition of our work, from problem diagnosis to method design. We would like to further clarify the main contributions of this paper as follows:
> > > 1. We identify a fundamental and previously overlooked limitation of mainstream Transformer routing design in the continual-learning setting with respect to forgetting. Based on this finding, we develop a rigorous theoretical framework that formalizes the connection between this limitation and forgetting susceptibility.
> > > 2. This insight directly motivates the design of an effective yet simple method for mitigating catastrophic forgetting. From this perspective, we believe that methodological simplicity is a virtue rather than a limitation. We evaluate the proposed approach on Qwen3-8B, Llama-3.2-3B, Ministral-3-8B, and Phi-4-mini-Instruct, and demonstrate its strong efficacy across multiple model families.
> > >
> > > **A2:** The comparison scope has been expanded to include MoLE (2024), MixLoRA (2024), and LoRA-Mixer (2025), together with OPLoRA (2026) [1] as a newer LoRA-based baseline. For methods not originally proposed for continual learning, a unified sequential protocol was used so that all methods can be compared under the same OP/BWT evaluation. Under this expanded comparison set, MH-MoE achieves the best OP/BWT tradeoff on both Llama-3.2-3B and Ministral-3-8B.
> > >
> > > |Model|Method|CS|FM|MB|PY|SQ|NC|ND|20M|OP|BWT|
> > > |---|---|---:|---:|---:|---:|---:|---:|---:|---:|---:|---:|
> > > |Llama-3.2-3B|SeqLoRA|42.3|45.0|23.0|40.3|66.7|54.3|50.8|42.9|45.7|-7.2|
> > > ||LoRAMoE|40.5|52.4|23.2|41.4|63.5|56.4|47.4|43.3|46.0|-7.1|
> > > ||MH-MoE|50.1|56.5|24.1|41.4|77.3|69.2|55.1|43.5|52.2|-2.2|
> > > ||MoLE|34.7|25.0|21.4|29.2|67.1|38.3|40.9|43.3|37.5|-11.5|
> > > ||LoRA-Mixer|38.6|42.7|21.5|39.5|65.9|35.8|42.5|43.5|41.3|-8.6|
> > > ||MixLoRA|41.5|47.2|21.7|42.6|70.7|53.1|30.1|43.3|43.8|-9.3|
> > > ||OPLoRA|43.8|56.1|23.4|41.9|69.1|50.6|56.3|43.2|48.1|-5.1|
> > > |Ministral-3-8B|SeqLoRA|44.6|60.3|18.3|44.8|66.4|56.7|69.5|43.0|50.5|-8.2|
> > > ||LoRAMoE|46.9|59.5|18.3|45.6|77.6|57.9|71.7|44.3|52.7|-11|
> > > ||MH-MoE|52.8|70.2|24.9|47.1|81.2|70.4|77.5|44.5|58.6|-1.7|
> > > ||MoLE|34.7|23.8|12.3|35.3|61.8|58.0|61.8|44.2|41.5|-15.6|
> > > ||LoRA-Mixer|34.8|26.2|13.6|37.4|56.1|56.8|61.2|44.1|41.3|-16.0|
> > > ||MixLoRA|48.9|53.6|13.0|44.5|77.7|70.4|62.5|42.7|51.7|-8.5|
> > > ||OPLoRA|51.5|65.3|23.6|45.9|74.1|72.8|74.8|42.3|56.3|-3.9|
> > >
> > > **A3:** Qwen3 results are already included in the original submission (see lines 317-320 and Table 1). In addition, the table in A2 provides results on newer mainstream backbones, including Llama-3.2-3B and Ministral-3-8B. Under the same sequential continual-learning protocol, MH-MoE continues to achieve higher OP and less negative BWT than the compared baselines on these newer models as well.
> > >
> > > **A4:** Thank you for this insightful comment. Lemma 2.1 is intended to provide the mixing-mass bound used in Theorem 2.2, while Theorem 2.2 offers a conditional susceptibility result: if compositions outside a small well-aligned subset are more likely to increase old-task loss after an update, then larger route-wise mixing, measured by $N_{\mathrm{eff}}(r)$, implies greater forgetting susceptibility. The assumptions are motivated by the empirical findings in Sections 2.1–2.2, which show that router inputs mix multiple feature compositions, with gradients relatively coherent within compositions but weakly aligned across them.
> > >
> > > The theory therefore is not intended to derive the full forgetting mechanism from first principles. Rather, it provides a unified mathematical framework linking the diagnosed mechanism, the metric $N_{\mathrm{eff}}(r)$, and route-level old-task loss increase. This gives formal support for the design motivation of MH-MoE: reducing composition mixing should reduce forgetting susceptibility. The revision will also discuss the limitations of the theory.
> > >
> > > Theorem 2.2 is a local, update-level result, consistent with prior continual-learning theory based on local loss approximations or update interference [2]. To complement this limitation of local analysis, additional multi-step empirical results were examined in the sequential setting. Across the training trajectory, MH-MoE consistently maintains higher OP and less negative BWT than LoRAMoE (standard MoE) after each task, providing complementary evidence that the local susceptibility characterized by the theorem remains relevant in the multi-step continual-learning process.
> > >
> > > | Stage | OP (LoRAMoE) | OP (MH-MoE) | BWT (LoRAMoE) | BWT (MH-MoE) |
> > > |---|---:|---:|---:|---:|
> > > | After CS  | 51.5 | 52.7 |  0.0 |  0.0 |
> > > | After FM  | 56.2 | 57.4 | -0.9 | -0.4 |
> > > | After MB  | 45.7 | 48.4 | -2.9 | -0.6 |
> > > | After PY  | 44.7 | 47.2 | -3.2 | -1.1 |
> > > | After SQ  | 49.9 | 53.4 | -3.8 | -1.4 |
> > > | After NC  | 50.2 | 52.3 | -5.2 | -1.9 |
> > > | After ND  | 48.4 | 52.8 | -6.1 | -2.1 |
> > > | After 20M | 46.0 | 52.2 | -7.1 | -2.2 |
> > >
> > > [1] Xiong and Xie, “OPLoRA”, AAAI, 2026.
> > > [2] Lanzillotta et al., “Local vs Global Continual Learning”, arXiv, 2024.

---

### Official Review · Reviewer_252v · 2026-02-28

**Soundness:** 3
**Presentation:** 2
**Significance:** 3
**Originality:** 3
**Overall Recommendation:** 5
**Confidence:** 2

**Summary:**

This paper investigates the causes of catastrophic forgetting in Mixture-of-Experts (MoE) Transformers under continual learning settings. The authors identify a key bottleneck: the post-attention representation, which serves as input to the router, is head-mixed and multi-feature, causing the router to respond to feature co-occurrences rather than separable signals. This leads to composition collisions, where different semantic/structural features are forced to share the same expert update destinations, exacerbating gradient conflicts and forgetting.

To address this, the authors propose MH-MoE (Multi-Head Mixture-of-Experts), which partitions the post-attention representation into head-aligned slices and performs independent routing per head with private expert banks. This design reduces feature composition collisions and mitigates forgetting. Experiments on the TRACE benchmark with Qwen3-0.6B/8B show consistent improvements in Overall Performance (OP) and Backward Transfer (BWT) over strong baselines like SeqLoRA, LoRAMoE, EWC, GEM, and O-LoRA.

**Compliance With Llm Reviewing Policy:**

Affirmed.

**Final Justification:**

My questions have been addressed, so I remain positive about this paper.

**Key Questions For Authors:**

- In Table 2, the gains appear smaller on the 8B model compared to the 0.6B model. Do the authors expect the benefit to diminish further at larger scales?

**Limitations:**

Yes.

**Strengths And Weaknesses:**

**Strengths**

* In Section 2, the authors combine empirical experiments with theoretical analysis to show that router inputs are **multiplexed and head-structured**, and that gradient updates can be perturbed by superposed feature compositions. This provides a plausible explanation for catastrophic forgetting in MoE and supports attributing it to the multi-head router structure.

* The proposed **effective composition number** $N_{eff}$ offers an intuitive metric for quantifying route-level feature mixing. The observed correlation between $N_{eff}$ and forgetting provides useful insight into when MoE routing fails to isolate learning signals.

* The **MH-MoE** architecture is simple and easily integrated into existing MoE systems. Experiments show consistent improvements in backward transfer on TRACE with minimal computational overhead, indicating a favorable performance–cost trade-off.

---

**Weaknesses**

* In Section 2, the paper states: “We instantiate $Y$ using salient semantic and structural variables: domain identity, stance label, token-frequency bucket, and relative-position bucket.” However, it is unclear how $Y$ is formally instantiated from these variables. The paper does not provide a sufficient explanation of how these variables are defined, extracted, or combined into the variable $Y$. Additional clarification on the construction of $Y$ and the role each variable plays in the analysis would significantly improve the clarity of the framework.

* In Lemma 2.1, the authors do not specify the range or assumptions on the quantity $\frac{m}{N_{eff}}$. If $\frac{m}{N_{eff}} \ge 1$ always holds, then the lemma becomes almost trivial and offers limited theoretical insight. Providing explicit assumptions or a discussion of the typical regime in which $\frac{m}{N_{eff}}$ lies would help readers better understand the significance and implications of the result.

* The experimental evaluation is primarily conducted on the TRACE benchmark using Qwen3-0.6B and Qwen3-8B models. While the results are promising, the robustness and generality of the proposed method remain unclear. Additional experiments on different benchmarks, model architectures, or scales would help establish whether the proposed approach consistently improves continual learning performance across diverse settings.

---

> ### Author Rebuttal · Authors · 2026-03-30
>
> We thank the reviewer for the careful reading and constructive comments. We address the questions on the construction of $Y$, the regime of Lemma 2.1, and robustness/scaling below.
>
> **W1 (Construction of $Y$):** We agree that the construction of $Y$ should be made explicit. In Section 2.1, $Y$ is defined as a discrete feature-composition variable whose value is linearly decodable from the post-attention representation. All analyses in Section 2 are conducted on C-STANCE and FOMC, and $Y$ is instantiated using four variables: domain identity, stance label, token-frequency bucket, and relative-position bucket. Domain identity is the dataset identity; stance label is the ground-truth example label assigned to the token representations within that example; token frequency is computed from the empirical token distribution over the two datasets and discretized into 6 bins; and relative position is defined by $p=\lfloor B\cdot t/L\rfloor$, where $t$ is token index, $L$ is sequence length, and $B=10$ is the number of buckets. We then define
> $Y=(\text{domain},\text{stance},\text{freq-bin},\text{pos-bin})$,
> so that each token is assigned a discrete composition label given by this tuple. We will revise the manuscript to make this construction and the role of each variable explicit.
>
> **W2 (Regime of Lemma 2.1):** We agree that the informative regime of Lemma 2.1 should be stated explicitly. The key quantity is $a:=\sqrt{m/N_{\mathrm{eff}}(r)}$, and the lemma gives
> $\Pr[C\in S]\le\sqrt{m/N_{\mathrm{eff}}(r)}$
> for any $|S|\le m$.
> Therefore, the meaningful regime is $m\le N_{\mathrm{eff}}(r)$, equivalently $a\le 1$; if $m>N_{\mathrm{eff}}(r)$, the bound becomes loose. Our argument is precisely about the former case, where $m$ denotes a small protected subset of compositions while $N_{\mathrm{eff}}(r)$ is large because the route mixes many compositions. This is also the empirically relevant regime: in the experiments, the measured $N_{\mathrm{eff}}$ values for LoRAMoE lie between 64.90 and 270.42 across all eight TRACE tasks, so for the small $m$ considered in the lemma, $m/N_{\mathrm{eff}}(r)$ is well below 1 in practice. We will revise the manuscript to state this regime explicitly and clarify why the lemma is informative in our setting.
>
>
> **W3 (Robustness across backbone families):** To validate robustness and generality, additional experiments were conducted on Phi-4-mini-Instruct and Llama-2-7B-Instruct. In both cases, MH-MoE again outperforms LoRAMoE, indicating that the benefit is not tied to a single backbone family.
>
> |Model|Method|CS|FM|MB|PY|SQ|NC|ND|20M|OP|BWT|
> |---|---|---:|---:|---:|---:|---:|---:|---:|---:|---:|---:|
> |Llama-2-7B-Instruct|SeqLoRA|24.2|50.2|18.7|31.6|60.5|34.6|26.5|43.2|36.2|-9.7|
> ||LoRAMoE|24.7|54.8|20.4|31.8|60.8|34.6|28.6|44.0|37.5|-9.3|
> ||MH-MoE|46.2|61.9|21.6|32.3|64.4|38.3|33.8|43.6|42.8|-4.1|
> |Phi-4-mini-Instruct|SeqLoRA|38.6|54.6|7.2|31.8|64.9|56.8|32.3|43.0|41.2|-11.6|
> ||LoRAMoE|38.2|55.6|9.3|32.1|65.8|60.4|31.1|42.6|41.9|-11.1|
> ||MH-MoE|49.3|68.3|9.5|32.9|77.9|65.4|40.6|43.2|48.4|-6.7|
>
> **Q1 (Scaling to larger models):** We further examined the smaller performance gain observed in the Qwen3-8B case. To better understand this behavior, the same comparison was repeated on Phi-4-mini-Instruct and Llama-2-7B-Instruct. As shown in W3, MH-MoE consistently achieves approximately a 5% BWT improvement over LoRAMoE, which is comparable to the 7% gain observed in the Qwen3-0.6B setting. These results suggest that the benefit of MH-MoE does not diminish monotonically with model scale. A plausible interpretation is that TRACE induces substantially milder forgetting on Qwen3-8B for all methods, leaving less room for absolute improvement on this benchmark.

---

> > ### Author Rebuttal · Reviewer_252v · 2026-04-03
> >
> > Thanks for your detailed response and the additional experiments. I will maintain my score.

---

> > > ### Author Response · Authors · 2026-04-08
> > >
> > > Thank you for your thoughtful follow-up and for taking the time to review the response carefully.

---

### Official Review · Reviewer_YbJ3 · 2026-03-11

**Soundness:** 3
**Presentation:** 3
**Significance:** 3
**Originality:** 3
**Overall Recommendation:** 4
**Confidence:** 4

**Summary:**

The paper investigates the causes of catastrophic forgetting in Mixture of Experts (MoE) Transformers within the context of continual learning. The authors argue that the standard architecture, where multi-head attention (MHA) outputs are concatenated before being fed into a router, creates a bottleneck. This design forces the router to handle complex, mixed feature compositions rather than individual, separable signals. Through a series of analyses, the paper demonstrates that these mixed inputs lead to increased interference within experts. To address this, the authors propose MH-MoE, a modified architecture that routes sub-representations from individual attention heads to their own sets of experts. Experiments on several continual learning benchmarks suggest that this fine-grained routing improves performance and reduces forgetting compared to standard MoE and other baseline methods.

**Compliance With Llm Reviewing Policy:**

Affirmed.

**Key Questions For Authors:**

- How does the model perform if you allow heads to share a global pool of experts but still route based on head-specific features? This would help clarify if the benefit comes from the "head-specific routing" or the "private expert capacity."

**Limitations:**

yes

**Strengths And Weaknesses:**

Strengths:
- The paper provides a very clear and intuitive explanation for why MoEs still suffer from forgetting despite their inherent sparsity. Focusing on the MHA-to-router interface is a fresh perspective compared to most works that only look at the router's loss or load balancing.
- The proposed MH-MoE is relatively simple to implement and does not require task-specific information or complex replay buffers, which makes it practical for real-world streaming scenarios.
- The performance gains on the TRACE benchmark are solid, particularly in reducing the backward transfer (forgetting) metric.

Weaknesses:
- A significant concern is the computational overhead of the proposed MH-MoE. While the paper mentions it is efficient, having multiple routers (one for each head) and potentially many more small experts could increase the kernel launch overhead and memory fragmentation on GPUs. The paper lacks a detailed latency or TFLOPS comparison against a standard MoE with the same parameter count.
- The scale of the experiments is somewhat limited. While Qwen3-0.6B is a reasonable starting point, modern MoE discussions often revolve around much larger scales where routing dynamics change. It is unclear if the "head-aligned" routing benefit holds as the number of heads and experts scales up significantly.

---

> ### Author Rebuttal · Authors · 2026-03-30
>
> We thank the reviewer for the careful reading and constructive comments.
>
> **W1 (Efficiency overhead):** In the current submission, Table 5 shows that the training-time overhead is about 1%: on Qwen3-8B, MH-MoE (M=8) achieves 3184.7 tok/s versus 3215.4 tok/s for LoRAMoE, with 160.85 ms/step versus 158.99 ms/step, and 41.95 GiB versus 40.22 GiB peak memory.
>
> To further quantify the efficiency trade-off, additional end-to-end decoding measurements were collected under the same evaluation setup:
>
> |Method|Tok/s|ms/tok|Mem(GB)|TFLOPS|
> |---|---:|---:|---:|---:|
> |LoRAMoE|62.49|16.00|17.49|0.786|
> |MH-MoE (M=8)|59.62|16.77|17.51|0.750|
> |MH-MoE (M=16)|57.40|17.42|18.12|0.722|
>
> Relative to LoRAMoE, MH-MoE (M=8) shows a 4.6% throughput drop and a 4.8% latency increase, with nearly unchanged peak memory. With M=16, the overhead becomes larger but remains moderate. Overall, these results indicate that MH-MoE does introduce measurable overhead, especially as M increases, but the degradation is moderate. We will add these latency, TFLOPS, and memory comparisons in the revision to make the efficiency trade-off more explicit.
>
> For transparency, the training-time results in Table 5 were measured on NVIDIA H100 GPUs, whereas the additional decoding measurements above were obtained on PPU-ZW810 due to time and compute constraints during the rebuttal period. Therefore, the absolute values should not be compared across the two platforms; only the within-platform comparisons are intended to be directly comparable. In the revision, these measurements will be consolidated under a unified hardware setup.
>
> **W2 (Scaling of the method):** The current submission includes results at two substantially different backbone scales, Qwen3-0.6B and Qwen3-8B, and MH-MoE outperforms LoRAMoE at both scales. To further test generality beyond the Qwen3 family, additional experiments were conducted on Phi-4-mini-Instruct and Llama-2-7B-Instruct, where MH-MoE again shows better retention than LoRAMoE (full results are given below in Q1). This indicates that the advantage is not specific to a single backbone family or to the 0.6B scale.
>
> In addition, larger routing configurations were evaluated on Llama-2-7B-Instruct by increasing both the number of experts and the number of routing heads, where $m$ denotes the number of routing heads and $e$ denotes the number of experts per head. Specifically, when the expert count is increased from 4 to 16, and when the head count is increased from 16 to 32, MH-MoE continues to outperform LoRAMoE under the corresponding matched settings. These results suggest that the benefit of head-aligned routing persists not only across larger backbones, but also as the routing architecture itself scales.
>
> |Scale axis|Method|Setting|OP|BWT|
> |---|---|---|---:|---:|
> |Expert scaling|LoRAMoE|e=4|37.5|-9.3|
> ||LoRAMoE|e=16|39.1|-6.9|
> ||MH-MoE|m=16, e=4|42.8|-4.1|
> ||MH-MoE|m=16, e=16|44.3|-2.2|
> |Head scaling|LoRAMoE|e=4|37.5|-9.3|
> ||MH-MoE|m=16, e=4|42.8|-4.1|
> ||MH-MoE|m=32, e=4|44.3|-2.8|
>
> **Q1 (Routing vs. private expert capacity):** To disentangle the effect of head-specific routing from that of head-private expert capacity, an additional variant was evaluated on Llama-2-7B-Instruct and Phi-4-mini-Instruct in which routing remains head-specific, but all heads share a global expert pool instead of using private experts.
>
> |Model|Method|CS|FM|MB|PY|SQ|NC|ND|20M|OP|BWT|
> |---|---|---:|---:|---:|---:|---:|---:|---:|---:|---:|---:|
> |Llama-2-7B-Instruct|SeqLoRA|24.2|50.2|18.7|31.6|60.5|34.6|26.5|43.2|36.2|-9.7|
> ||LoRAMoE|24.7|54.8|20.4|31.8|60.8|34.6|28.6|44.0|37.5|-9.3|
> ||MH-MoE|46.2|61.9|21.6|32.3|64.4|38.3|33.8|43.6|42.8|-4.1|
> ||MH-MoE (shared expert pool)|41.6|63.3|20.9|31.3|61.5|37.0|30.2|43.6|41.2|-5.6|
> |Phi-4-mini-Instruct|SeqLoRA|38.6|54.6|7.2|31.8|64.9|56.8|32.3|43.0|41.2|-11.6|
> ||LoRAMoE|38.2|55.6|9.3|32.1|65.8|60.4|31.1|42.6|41.9|-11.1|
> ||MH-MoE|49.3|68.3|9.5|32.9|77.9|65.4|40.6|43.2|48.4|-6.7|
> ||MH-MoE (shared expert pool)|44.5|63.5|9.5|32.4|79.9|64.2|36.9|43.0|46.7|-8.0|
>
> This control indicates that the gain does not come solely from private expert capacity. Even with a shared global expert pool, head-specific routing already improves substantially over standard LoRAMoE, indicating that routing on less mixed subrepresentations is itself an important contributor. At the same time, full MH-MoE performs better than the shared-pool variant on both backbones, showing that head-private experts provide an additional benefit.
>
> Our interpretation is therefore that both components matter: head-specific routing is the primary source of the improvement, while private expert capacity further strengthens specialization by reducing competition across heads for the same expert parameters. We will add this ablation and discussion in the revision.

---

> > ### Author Rebuttal · Reviewer_YbJ3 · 2026-04-03
> >
> > My concerns have been addressed.

---

> > > ### Author Response · Authors · 2026-04-08
> > >
> > > Thank you for your follow-up and for reviewing our response. We are glad that our response addressed your concerns.

---

### Official Review · Reviewer_hvZL · 2026-03-12

**Soundness:** 3
**Presentation:** 3
**Significance:** 3
**Originality:** 3
**Overall Recommendation:** 4
**Confidence:** 4

**Summary:**

This manuscript investigates why MoE Transformers still suffer catastrophic forgetting despite sparse routing. The authors argue the bottleneck lies before routing: multi-head attention concatenates head-specific signals into a single vector, forcing the router to act on mixed feature compositions. They formalize this via a route-wise effective composition number N_eff, prove theoretically that higher N_eff increases forgetting susceptibility, and propose MH-MoE, which routes independently over H head-aligned slices of the post-attention representation using head-private expert banks. Experiments on TRACE with Qwen3-0.6B/8B show consistent improvements in BWT and overall performance over LoRAMoE and other continual learning baselines at minimal computational cost.

**Compliance With Llm Reviewing Policy:**

Affirmed.

**Final Justification:**

The clarifications provided have resolved my original questions, and I will maintain my score of 4.

**Key Questions For Authors:**

- Have the authors measured N_eff using the actual H=8 uniform slices applied in MH-MoE, rather than attention-head-aligned blocks? If composition mixing under MH-MoE's own slicing is not demonstrably lower than under h_t, the mechanistic account needs revision.
- Why does M=4 underperform M=2 in Table 4? What is the computational overhead of M=16, and why was M=8 chosen over the better-performing M=16 for the main experiments?
- How do N_eff values and gradient alignment patterns look on the six TRACE tasks excluded from the analysis, particularly for tasks where MH-MoE and LoRAMoE perform similarly?
- The Qwen3-8B improvement is marginal. Do the authors expect MH-MoE's benefits to diminish further at larger model scales, and if so, what does this imply for practical applicability?

**Limitations:**

Please see the weaknesses and questions section for further detail.

**Strengths And Weaknesses:**

Strengths
- The combination of linear probing, gradient direction analysis, and N_eff measurement builds a coherent multi-level argument connecting head-mixed representations to forgetting, rather than simply reporting empirical gains.
- Task-agnostic and low-overhead. MH-MoE requires no task boundaries or replay buffers and adds only ~1% latency and ~4% memory overhead, making it practical for streaming continual learning.

Weaknesses
- The diagnosis is framed in terms of attention head structure, but MH-MoE slices h_t by contiguous dimension intervals after O-projection. Since W_O is an unconstrained d×d linear map, it generally mixes all head outputs across all output dimensions, so contiguous slices of h_t need not correspond to individual head channels. The head ablation experiments (Fig. 2/7) implicitly assume O-projection preserves head-aligned structure—an assumption that is neither justified nor verified. A permuted-slice baseline would directly test this.
- All probing and gradient analyses are performed on only two of the eight TRACE tasks (C-STANCE and FOMC), which share a stance detection structure. Whether the same composition mixing patterns hold for structurally different tasks such as Py150 or 20Minuten—where MH-MoE's gains are marginal—is not verified.
- M=16 outperforms M=8 yet is not used in the main experiments, with no overhead numbers reported for M=16. More critically, M=4 underperforms M=2, which directly contradicts the paper's narrative that more routing heads reduce composition mixing and improve retention.
- All experiments use a single benchmark (TRACE) and a single model family (Qwen3). Gains on Qwen3-8B are small (BWT −5.5% → −5.1%), raising questions about whether benefits diminish at larger scales.

---

> ### Author Rebuttal · Authors · 2026-03-30
>
> We thank the reviewer for the careful reading and detailed technical questions.
>
> **W1 / Q1 (Actual slicing used by MH-MoE):** The $N_{\mathrm{eff}}$ results in Fig. 6 are measured on the actual contiguous post-$W_O$ slices used by MH-MoE, rather than on pre-$W_O$ attention-head blocks. Thus, the mechanistic analysis is defined on the same representation and slicing scheme used by the method itself. Under this actual slicing, MH-MoE yields substantially lower $N_{\mathrm{eff}}$ than LoRAMoE, so the reduction in composition mixing is verified on the representation where MH-MoE operates. We agree that the terminology should be clearer: the “heads” in MH-MoE are routing slices of the post-attention representation after $W_O$, rather than literal preserved pre-$W_O$ attention heads.
>
> Additional results of the permuted-slice control are shown below. MH-MoE with permuted-slices performs much worse than MH-MoE on both Llama-2-7B-Instruct and Phi-4-mini-Instruct, showing that the gain does not come from arbitrary partitioning. We will revise the manuscript accordingly.
>
>
> |Model|Method|CS|FM|MB|PY|SQ|NC|ND|20M|OP|BWT|
> |---|---|---:|---:|---:|---:|---:|---:|---:|---:|---:|---:|
> |Llama-2-7B-Instruct|MH-MoE|46.2|61.9|21.6|32.3|64.4|38.3|33.8|43.6|42.8|-4.1|
> ||Perm-slice|31.0|32.3|19.3|25.9|57.9|4.9|0.0|42.1|26.7|-16.7|
> |Phi-4-mini-Instruct|MH-MoE|49.3|68.3|9.5|32.9|77.9|65.4|40.6|43.2|48.4|-6.7|
> ||Perm-slice|43.3|23.0|7.2|23.7|74.1|0.0|0.0|42.8|26.8|-26.7|
>
> **W2 / Q3 (All-task analysis across TRACE):** Results of further analysis across all eight TRACE tasks are shown below. Across all tasks, within-composition gradient alignment remains higher than across-composition alignment, indicating that the composition-separation phenomenon is not limited to C-STANCE and FOMC. At the same time, its strength is task-dependent: the within-minus-across gap is larger on ScienceQA, NumGLUE-ds, NumGLUE-cm, FOMC, and C-STANCE, but smaller on Py150, MeetingBank, and 20Minuten, where MH-MoE and LoRAMoE are also closer. We also computed $N_{\mathrm{eff}}$ on all eight tasks and found that MH-MoE consistently yields much smaller $N_{\mathrm{eff}}$ than LoRAMoE throughout TRACE. We will include the full task-wise results in the revision.
>
>
> **Gradient alignment across all 8 TRACE tasks**
>
> |Task|Within|Across|Gap|
> |---|---:|---:|---:|
> |ScienceQA|0.683|0.026|0.657|
> |Py150|0.288|0.021|0.268|
> |NumGLUE-ds|0.907|0.076|0.831|
> |NumGLUE-cm|0.754|0.070|0.684|
> |MeetingBank|0.300|0.016|0.284|
> |FOMC|0.741|0.057|0.684|
> |C-STANCE|0.606|0.059|0.546|
> |20Minuten|0.364|0.028|0.336|
>
> **$N_{\mathrm{eff}}$ comparison**
>
> |Task|LoRAMoE|MH-MoE|
> |---|---:|---:|
> |ScienceQA|269.09|10.74|
> |Py150|132.49|9.50|
> |NumGLUE-ds|103.41|3.77|
> |NumGLUE-cm|64.90|3.35|
> |MeetingBank|270.42|15.15|
> |FOMC|72.04|5.71|
> |C-STANCE|93.96|3.99|
> |20Minuten|238.59|5.00|
>
> **W3 / Q2 (Head count and efficiency):** The main results in Table 1 use $M=16$, which is also the best-performing setting in Table 4. Thus, the headline results are reported with the strongest MH-MoE configuration we tested, rather than with $M=8$. Regarding why $M=4$ underperforms $M=2$, we do not interpret Table 4 as implying a monotonic dependence on $M$ at small head counts. Increasing $M$ improves routing granularity, but also narrows each slice ($d/M$), so each router makes decisions from less information. The $M=2$ to $M=4$ result suggests that, in this regime, the latter effect can slightly outweigh the former. Since performance improves again at $M=8$ and is best at $M=16$, we view Table 4 as reflecting a non-monotonic trade-off at small head counts rather than contradicting the overall benefit of finer routing. We use $M=8$ only in the controlled analysis of Section 5.3 to match route-space size against standard LoRAMoE: $4^8=65536$ is close to $\binom{26}{5}=65780$, whereas $4^{16}\approx 4.3\times10^9$ would make such a matched comparison impractical.
>
> In decoding-time efficiency, $M=16$ incurs only moderate overhead relative to $M=8$:
>
> |Method|Tok/s|ms/tok|Mem(GB)|TFLOPS|
> |---|---:|---:|---:|---:|
> |LoRAMoE|62.49|16.00|17.49|0.786|
> |MH-MoE (M=8)|59.62|16.77|17.51|0.750|
> |MH-MoE (M=16)|57.40|17.42|18.12|0.722|
>
> Hardware details are given in W1 for Reviewer YbJ3 (sorry for the 5000-character limit).
>
> **W4 / Q4 (Scaling to larger backbones):** We further examined the smaller performance gain observed in the Qwen3-8B case. To better understand this behavior, the same comparison was repeated on Phi-4-mini-Instruct and Llama-2-7B-Instruct. As shown in W3 of Reviewer 252v (sorry for the 5000-character limit), MH-MoE consistently achieves approximately a 5% BWT improvement over LoRAMoE, which is comparable to the 7% gain observed in the Qwen3-0.6B setting. These results suggest that the benefit of MH-MoE does not diminish monotonically with model scale. A plausible interpretation is that TRACE induces substantially milder forgetting on Qwen3-8B for all methods, leaving less room for absolute improvement on this benchmark.

---

> > ### Author Rebuttal · Reviewer_hvZL · 2026-04-04
> >
> > Thank you for the rebuttal. The clarifications provided have resolved my original questions, and I will maintain my score of 4.

---

> > > ### Author Response · Authors · 2026-04-08
> > >
> > > Thank you for the thoughtful follow-up and for confirming that the rebuttal addressed your original questions.

---

### Decision · Program_Chairs · 2026-04-30

**Decision:**

Accept (regular)

**Comment:**

This paper proposes MH-MoE, a framework that identifies a pre-routing bottleneck in MoE Transformers and mitigates catastrophic forgetting via head-wise routing over sub-representations.

Reviewers appreciate its
- clear diagnosis-to-design pipeline connecting mechanism analysis with architectural improvement,
- lightweight design with consistent gains in BWT across settings.

However, concerns focus on
- limited evaluation scope, mainly on TRACE and a restricted set of models,
- insufficient clarity and rigor in parts of the theoretical formulation and assumptions,
- potential computational overhead and efficiency trade-offs,
- questions on novelty and comparison with more recent LoRA-MoE variants.

The rebuttal provides additional experiments, expanded baselines, full-task analyses, and detailed efficiency measurements, which largely address the reviewers’ concerns. As reflected in the final justifications, most reviewers acknowledge that their concerns have been resolved and maintain or support acceptance.

For the final revision, the authors are encouraged to explicitly incorporate key rebuttal clarifications into the main paper, including
- broader benchmark evaluations,
- clearer theoretical assumptions,
- expanded experiments on other backbones than Qwen-3,
- expanded comparisons with recent methods,
- a more explicit discussion of limitations,
to further strengthen the paper’s clarity and impact.